# In situ continuous Dopa supply by responsive artificial enzyme for the treatment of Parkinson's disease

Xiao Fang [1], Meng Yuan[1], Fang Zhao[1], Aoling Yu[1], Qianying Lin[1], Shiqing Li[1], Huichen Li[1], Xinyang Wang[1], Yanbin Yu[1], Xin Wang[1], Qitian Lin[1], Chunhua Lu [1] ✉ & Huanghao Yang [1] ✉

Oral dihydroxyphenylalanine (Dopa) administration to replenish neuronal dopamine remains the most effective treatment for Parkinson's disease (PD). However, unlike the continuous and steady dopamine signaling in normal neurons, oral Dopa induces dramatic fluctuations in plasma Dopa levels, leading to Dopa-induced dyskinesia. Herein, we report a functional nucleic acid-based responsive artificial enzyme (FNA-$Fe_3O_4$) for in situ continuous Dopa production. FNA-$Fe_3O_4$ can cross the blood-brain barrier and target diseased neurons relying on transferrin receptor aptamer. Then, FNA-$Fe_3O_4$ responds to overexpressed α-synuclein mRNA in diseased neurons for antisense oligonucleotide treatment and fluorescence imaging, while converting to tyrosine aptamer-based artificial enzyme (Apt-$Fe_3O_4$) that mimics tyrosine hydroxylase for in situ continuous Dopa production. In vivo FNA-$Fe_3O_4$ treatment results in recovery of Dopa and dopamine levels and decrease of pathological overexpressed α-synuclein in PD mice model, thus ameliorating motor symptoms and memory deficits. The presented functional nucleic acid-based responsive artificial enzyme strategy provides a more neuron friendly approach for the diagnosis and treatment of PD.

Parkinson's disease (PD) is currently the second leading neurodegenerative disease in the world, causing severe movement impairments in patients[1]. The main pathological changes in PD are the degeneration and death of substantia nigra dopaminergic neurons[2], and the resulting significant decrease of dopamine content in striatum[3,4]. Oral dihydroxyphenylalanine (Dopa) administration is currently the most effective treatment for PD[5]. Dopa can cross the blood-brain barrier (BBB) and is converted to dopamine in neurons[6]. However, due to the short half-life of Dopa, Dopa administration can cause large fluctuations in plasma levels of Dopa, resulting in Dopa-induced dyskinesia[7,8]. Also, undesired metabolism of Dopa in peripheral tissues can cause adverse reactions[9,10].

The degeneration of neurons is accompanied by the inactivation of tyrosine hydroxylase[11,12], the rate-limiting enzyme that catalyzes the synthesis of catecholamine neurotransmitters, which is responsible for the conversion of tyrosine to Dopa[13]. Thus, an inspiration is that replacing the inactivated tyrosine hydroxylase in diseased neurons with nanozymes might enable sustained and steady in situ generation of Dopa[14]. Nanozyme is a class of nanomaterials with catalytic activity[15,16]. Strategies for alleviating oxidative stress based on catalase-like and superoxide dismutase-like nanozymes have been well-applied in neurodegenerative diseases[17–19]. It is reported that mononuclear nonheme iron is the active center of tyrosine hydroxylase, which carries out H atom abstraction followed by hydroxylation on aromatic ring through Fe(IV)=O intermediate[20,21]. Based

[1]MOE Key Laboratory for Analytical Science of Food Safety and Biology, Fujian Provincial Key Laboratory of Analysis and Detection Technology for Food Safety, State Key Laboratory of Photocatalysis on Energy and Environment, College of Chemistry, Fuzhou University, Fuzhou 350108, P. R. China. ✉e-mail: chunhualu@fzu.edu.cn; hhyang@fzu.edu.cn

on the investigation of tyrosine hydroxylase and the mastery of the properties of nanozymes with iron-reactive sites[22], we speculate that the formation of Fe(IV)=O intermediates mediated by $Fe_3O_4$ nanozymes would be a feasible alternative to tyrosine hydroxylase pathway[23–25]. In this study, we first explained the detailed catalysis of $Fe_3O_4$ nanoparticles on the hydroxylation of tyrosine to produce Dopa in the presence of hydrogen peroxide ($H_2O_2$) and ascorbic acid (AA) (Phase I in Fig. 1a). The well-defined tyrosine hydroxylase-mimicking activity of $Fe_3O_4$ nanozymes makes it potential to achieve continuous and steady Dopa supply.

Nanozymes can simulate the catalytic active sites of native enzymes, while it still lacks the function of substrate capture, which makes its intracellular catalytic rate hindered by the macromolecular crowding[26]. Aptamers, as substrate-specific binding components, have been fully acknowledged for their function of mimicking the recognition of biological enzymes[27,28]. Therefore, in this study, we further constructed the tyrosine aptamer-based artificial enzyme (Apt-$Fe_3O_4$), in which $Fe_3O_4$ as the catalytic active center and the tyrosine aptamer[29] as the substrate binding site (Phase II in Fig. 1a). The experimental results showed that Apt-$Fe_3O_4$ was found to retain superior catalytic

performance than $Fe_3O_4$ nanozymes even in the crowded environment. Therefore, we speculate that Apt-$Fe_3O_4$ with specificity and high efficiency could be a favorable alternative to intracellular tyrosine hydroxylase.

Two prerequisites are required for Apt-$Fe_3O_4$ to successfully function as tyrosine hydroxylase mimic in PD, namely the enrichment of Apt-$Fe_3O_4$ in brain neurons and the avoidance of the side effects of Dopa production on peripheral tissues[30,31]. Since both capillary endothelial cells and PD neurons express transferrin receptor (TfR), TfR aptamer-functionalized nanomaterials can cross the BBB and be efficiently internalized by diseased neurons via receptor-mediated transport pathway[32,33]. Furthermore, aberrantly overexpressed α-synuclein (SNCA) and its mRNA are regarded as important targets for the diagnosis and treatment of PD, because they are biomarkers and important causes of dopaminergic neuron death in PD[34]. The degenerated neurons-specific Dopa production could be realized through the responsiveness to SNCA mRNA by functional nucleic acid designs.

Here, we design the functional nucleic acid-based responsive artificial enzyme (FNA-$Fe_3O_4$) on the basis of Apt-$Fe_3O_4$. The functional

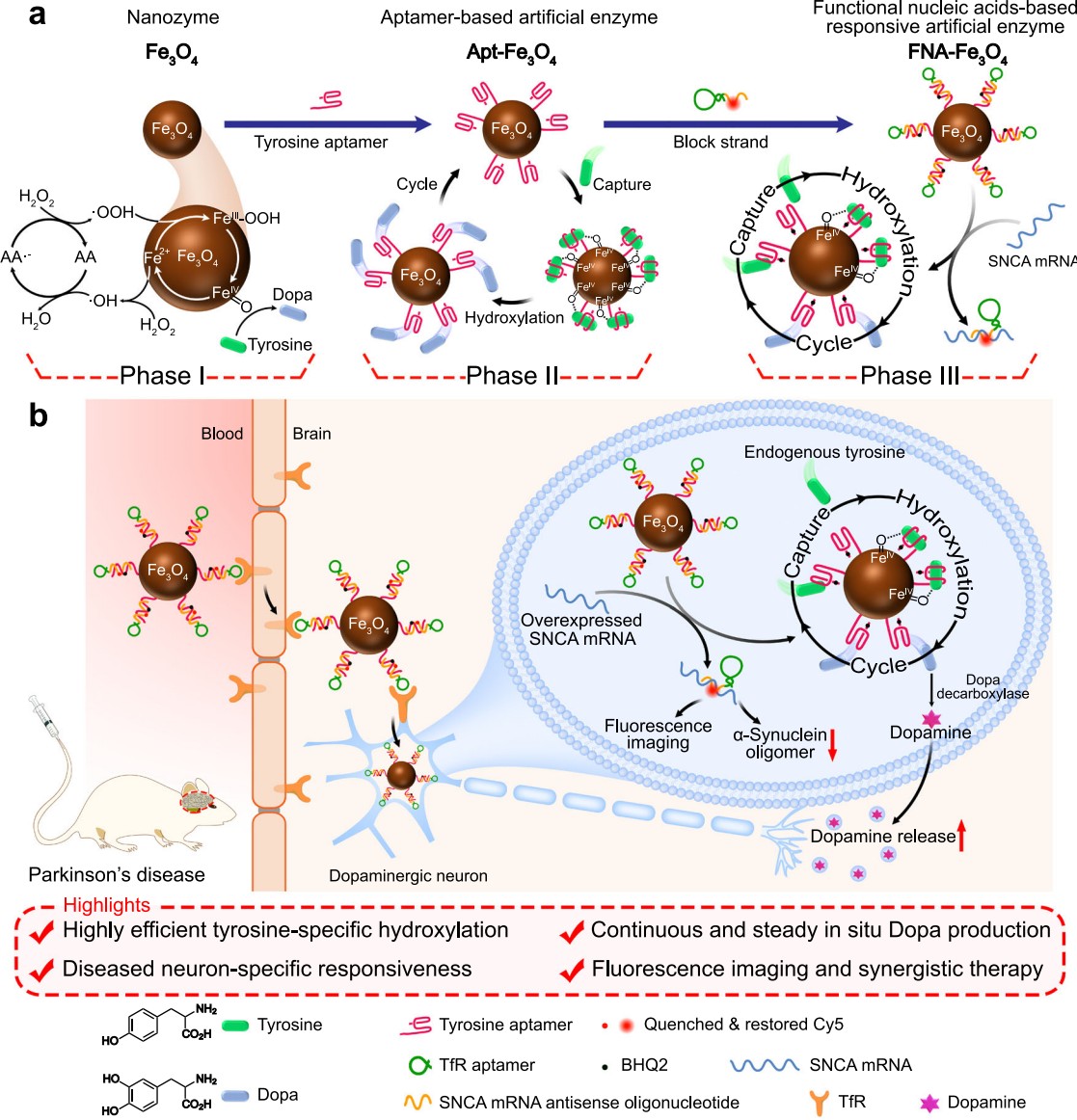

**Fig. 1 | Schematic diagram of FNA-Fe₃O₄ for the treatment of PD. a** Catalytic mechanism of Fe₃O₄ nanozymes on the hydroxylation of tyrosine to produce Dopa (Phase I), illustration of the functions of Apt-Fe₃O₄ (Phase II) and FNA-Fe₃O₄ (Phase III). **b** Illustration of FNA-Fe₃O₄ for synergistic treatment of PD.

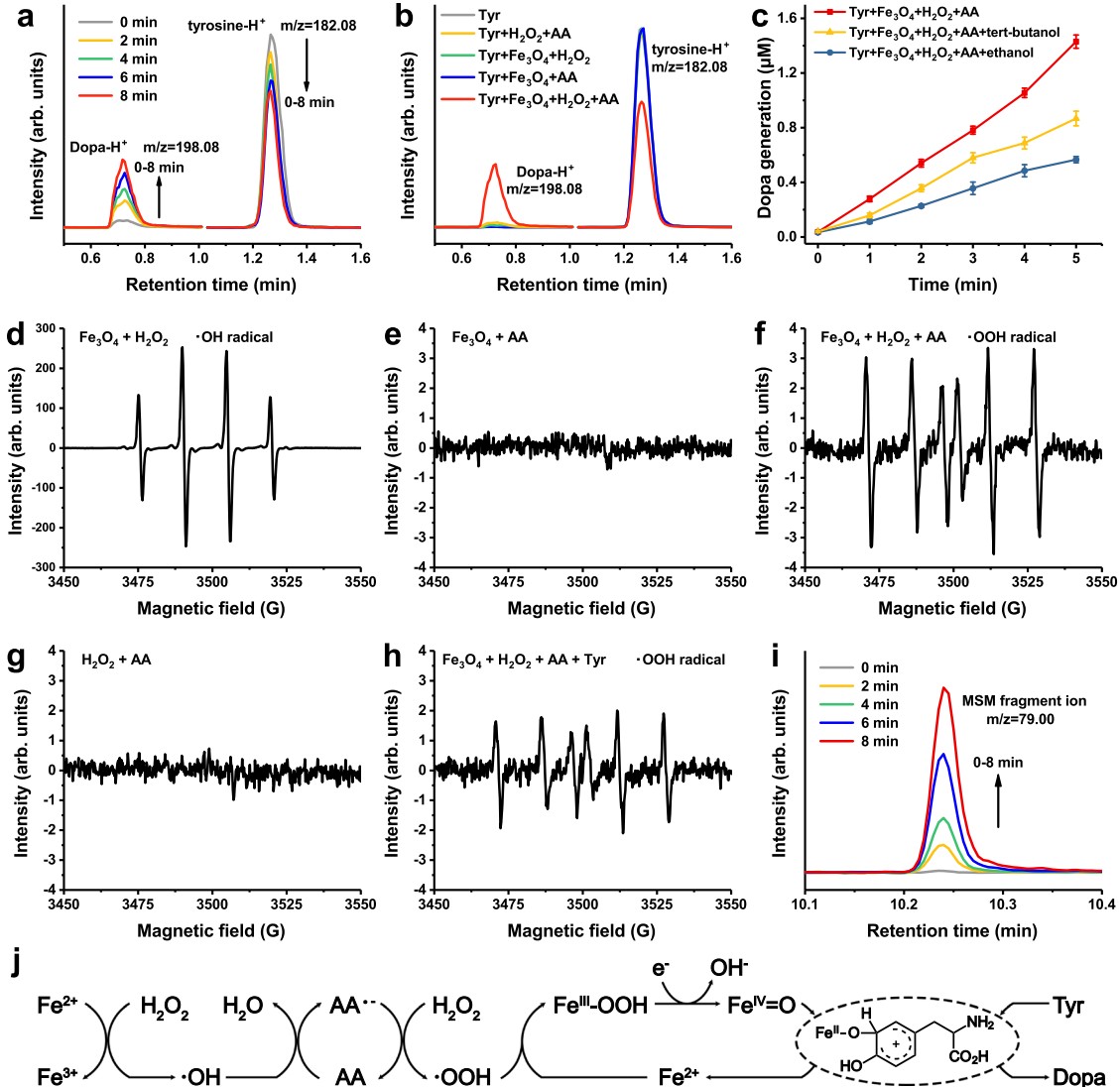

**Fig. 2 | The tyrosine hydroxylase-mimicking activity of Fe₃O₄ nanoparticles.**
**a** Time-dependent UPLC-MS peak intensities changes of reactant tyrosine and product Dopa in the presence of 10 µg mL⁻¹ Fe₃O₄ nanoparticles, 100 µM tyrosine, 5 mM H₂O₂, and 5 mM AA. Peaks at m/z = 182.08 and m/z = 198.08 represent tyrosine and Dopa respectively. **b** UPLC-MS peak intensities of reactant tyrosine and product Dopa in systems with different reaction conditions. **c** Dopa generation in the reaction system (10 µg mL⁻¹ Fe₃O₄ nanoparticles, 100 µM tyrosine, 5 mM H₂O₂ and 5 mM AA) in the presence of 1 mM tert-butanol or ethanol. The results were expressed as mean ± SD (*n* = 3 independent experiments). **d–h** ESR spectra corresponding to the ·OH generated by Fe₃O₄ nanoparticles in the presence of H₂O₂ (**d**), no free radical generated by Fe₃O₄ nanoparticles in the presence of AA (**e**), the ·OOH generated by Fe₃O₄ nanoparticles in the presence of H₂O₂ and AA (**f**), no free radical generated in the mixture of H₂O₂ and AA (**g**), the ·OOH generated by Fe₃O₄ nanoparticles in the presence of H₂O₂ and AA with added tyrosine (**h**). **i** Time-dependent GC-MS peak intensities of MSM in the presence of 10 µg mL⁻¹ Fe₃O₄ nanoparticles, 1 mM DMSO, 5 mM H₂O₂ and 5 mM AA. Peak at m/z = 79.00 represent MSM formation. **j** Schematic suggested mechanism for the production of Fe(IV) = O intermediate by Fe₃O₄ nanoparticles in the presence of H₂O₂ and AA and the catalytic process of hydroxylation of tyrosine to Dopa. **a**, **b**, **d–i** one representative data was shown from three independently repeated experiments. Source data are provided as a Source Data file.

nucleic acid structure is a double strand formed by the hybridization of tyrosine aptamer and a block strand containing SNCA mRNA antisense oligonucleotide (ASO)[35] and TfR aptamer[36]. FNA-Fe₃O₄ can potently penetrate the BBB and be enriched in neurons by virtue of the TfR aptamer. In diseased neurons, FNA-Fe₃O₄ is converted to Apt-Fe₃O₄ in response to high levels of SNCA mRNA to exhibit tyrosine hydroxylase-mimicking activity for Dopa production (Phase III in Fig. 1a). Meanwhile, strand displacement reactions trigger ASO therapy and fluorescence imaging of overexpressed SNCA mRNA. FNA-Fe₃O₄ integrates responsive tyrosine hydroxylase-mimicking activity, ASO therapy, and fluorescence imaging of PD, which avoids the dramatic fluctuation of dopamine caused by conventional Dopa administration, revealing the application prospects of functional nucleic acid-based responsive artificial enzymes in biomedicine.

## Results

### Fe₃O₄ mediated tyrosine hydroxylation

Fe₃O₄ nanoparticles were previously reported to catalyze the multi-step oxidation of tyrosine, passing through intermediate product Dopa to yield final product dopachrome[23]. However, the detailed mechanism of the formation of the intermediate product Dopa has not been explained. Thus, we explored the stable existence of the intermediate product Dopa, and the catalytic process of the tyrosine hydroxylation to produce Dopa by Fe₃O₄ in the presence of pathological concentrations of AA and H₂O₂[37,38]. Commercially available 15 nm water-dispersible surface carboxyl-modified Fe₃O₄ nanoparticles were utilized as the model (Supplementary Fig. 1), and then applied to examine the catalyzed hydroxylation of tyrosine to Dopa. Figure 2a showed the content changes during the hydroxylation of 100 µM

tyrosine to Dopa in the presence of 10 μg mL$^{-1}$ Fe$_3$O$_4$, 5 mM H$_2$O$_2$, and 5 mM AA, as tracked by time-dependent ultrahigh performance liquid chromatography-mass (UPLC-MS). The decrease of tyrosine and increase of Dopa indicated the gradual conversion of tyrosine to Dopa during the reaction. Each component in the reaction mixture was essential to drive the hydroxylation of tyrosine to Dopa (Fig. 2b). According to the reactivity of these components, there might be a variety of active radical intermediates. Tert-butanol tends to mainly quench ·OH, and ethanol is known to effectively scavenge both ·OH and Fe(IV)=O[39]. The addition of tert-butanol in the reaction system mildly inhibited the reaction by 39.4% after 5 min, while the addition of ethanol induced more serious inhibition by 60.4% (Fig. 2c). These differences could be attributed to the participation of ·OH and high-spin Fe(IV)=O in hydroxylation.

To further understand the mechanism that Fe$_3$O$_4$ catalyzed the hydroxylation of tyrosine to Dopa in the presence of H$_2$O$_2$ and AA, we performed a series of electron spin resonance (ESR) experiments and correlated these results with the active intermediates of the reaction. Figure 2d displayed a distinct hydroxyl radical (·OH) signal in the mixed system in which Fe$_3$O$_4$ coexisted with H$_2$O$_2$, which was consistent with the inherent peroxidase activity of Fe$_3$O$_4$ nanozymes[40]. Figure 2e showed that no detectable free radicals were generated when Fe$_3$O$_4$ coexisted with AA. Whereas, in the presence of Fe$_3$O$_4$, H$_2$O$_2$ and AA, a new free radical signal was observed as peroxy radical (·OOH), and this was accompanied by complete disappearance of ·OH in the system (Fig. 2f). No direct radical reaction occurred between H$_2$O$_2$ and AA (Fig. 2g). These results indicated that AA was a key promoter of ·OOH formation and led to ·OH depletion. Further, the addition of the tyrosine substrate to the mixed system of ·OOH resulted in a decrease in the intensity of ·OOH without the change of radical species (Fig. 2h), implying that ·OOH provided the key species for the transformation of tyrosine to Dopa. It is reported that the ·OOH combined with Fe$^{2+}$ to generate high-spin Fe(IV)=O intermediate[41], which had the same structure in analogy to the active center of biological protein tyrosine hydroxylase. Dimethyl sulfoxide (DMSO) was used as the probe of Fe(IV)=O, which readily react with Fe(IV)=O to generate dimethyl sulfone (MSM) through oxygen atom transfer[42]. According to the gas chromatography-mass (GC-MS) analysis results in Fig. 2i, the peak of MSM gradually increased with reaction time, suggesting the possible occurrence of Fe(IV)=O. These results provided a valid explanation for the mechanistic path of tyrosine hydroxylation (Fig. 2j). The ·OH produced by the reaction of Fe$_3$O$_4$ with H$_2$O$_2$ was converted into ·OOH under the catalysis of AA. Then ·OOH mediated the generation of the active intermediate Fe(IV)=O, and finally hydroxylated tyrosine to generate Dopa.

## Construction and characterizations of Apt-Fe$_3$O$_4$

The covalent linkage of aptamer binding sites to nanozymes is introduced as a versatile method to improve the catalytic activity of nanozymes by concentrating the reaction substrates at the catalytic nanozyme core[28], thereby emulating the binding and catalytic active-site functions of native enzymes (Fig. 3a). After confirming the tyrosine hydroxylase-mimicking activity, Fe$_3$O$_4$ nanozymes were covalently conjugated with the 5'-terminal amino-modified tyrosine aptamer through EDC/NHS mediated amide condensation reaction to construct the tyrosine aptamer-based artificial enzyme Apt-Fe$_3$O$_4$. We also prepared random sequence DNA-conjugated Fe$_3$O$_4$ (Ran-Fe$_3$O$_4$) for comparison (Supplementary Fig. 2). Then, the amount of DNA loaded on Apt-Fe$_3$O$_4$ through amide bonds was quantified as 4.59 μmol g$^{-1}$, corresponding to about 25 aptamer strands per nanoparticle (Supplementary Figs. 3, 4).

We examined the hydroxylation rates of tyrosine to Dopa at variable concentrations of tyrosine (in all experiments, 5 mM H$_2$O$_2$ and 5 mM AA were used) catalyzed by Fe$_3$O$_4$, Ran-Fe$_3$O$_4$ and Apt-Fe$_3$O$_4$ (10 μg mL$^{-1}$) (Supplementary Fig. 5), as a function of tyrosine

concentrations, outlined in the kinetic curve of Fig. 3b. For Apt-Fe$_3$O$_4$ or controls, Michaelis-Menten-type saturation kinetic curves were observed (kinetic parameters were listed in Supplementary Table 2). Compared with bare Fe$_3$O$_4$, the catalytic activity of Ran-Fe$_3$O$_4$ was almost the same, while Apt-Fe$_3$O$_4$ was found to possess a significantly higher catalytic activity. The enhanced catalytic activity of Apt-Fe$_3$O$_4$ was attributed to the high local concentration of the substrates close to the catalytic interface, resulting from the specific binding of aptamers to tyrosine substrates.

It is worth noting that the aqueous solution, as the simplest model, cannot simulate the actual intracellular macromolecular crowding environment[26]. Thus, the catalytic abilities of Apt-Fe$_3$O$_4$ and controls were evaluated in a simulated crowded environment with 20 wt% PEG-20000 as a crowding agent. As shown in Fig. 3c and Supplementary Fig. 6, the catalytic rates of Fe$_3$O$_4$ and Ran-Fe$_3$O$_4$ declined markedly compared to that in aqueous solution, implying that their catalytic functions were severely disturbed in the complex cellular environment. However, Apt-Fe$_3$O$_4$ satisfactorily retained most of its catalytic efficiency (kinetic parameters were listed in Supplementary Table 2), which was undoubtedly attributed to the selective capture of tyrosine by aptamers. Through the specific binding of tyrosine and aptamer, the residence time of tyrosine on the Apt-Fe$_3$O$_4$ was increased, which was beneficial to the formation of active intermediates, making the reaction easy to carry out in crowded environments. This feature was advantageous for tyrosine aptamer-based artificial enzyme Apt-Fe$_3$O$_4$ to perform catalytic assignments in the complex intracellular environment.

## Construction and characterizations of FNA-Fe$_3$O$_4$

Next, to achieve the efficient BBB crossing and neuron-specific responsiveness of the artificial enzyme for PD treatment, the block strand composed of SNCA antisense oligonucleotide and TfR aptamer was designed and hybridized to the tyrosine aptamer on Apt-Fe$_3$O$_4$ to construct the functional nucleic acid-based responsive artificial enzyme FNA-Fe$_3$O$_4$ (Fig. 3d and Supplementary Fig. 7). The dynamic light scattering result revealed that the hydrodynamic size of FNA-Fe$_3$O$_4$ was centralized around 50 nm (Supplementary Fig. 7e), which was suitable for BBB crossing[33,43].

Then we verified the strand displacement reactivity of FNA-Fe$_3$O$_4$ response to SNCA mRNA by polyacrylamide gel electrophoresis (PAGE). As shown in Fig. 3e, the block strand could first hybridize with the tyrosine aptamer (lane 4). This double strand was stable in physiological concentrations of tyrosine (lane 8). However, the block strand was subsequently displaced by SNCA mRNA to release free tyrosine aptamer (lane 10). Meanwhile, monitoring the content of SNCA related biomarkers can provide an effective means for fluorescence imaging of PD[44]. Therefore, the block strand and aptamer strand on FNA-Fe$_3$O$_4$ were modified with Cy5 and BHQ2, respectively, to form fluorescence-quenching pairs. The fluorescence-quenching pairs were separated after the SNCA mRNA triggered strand displacement reaction, thus generating turn-on fluorescence signal for in situ fluorescence imaging of abnormally overexpressed SNCA mRNA (Supplementary Fig. 7a). The fluorescent signal of Cy5 was significantly attenuated after hybridization of the block strand to the BHQ2-labeled aptamer strand, and then largely restored with the addition of target mRNA (Supplementary Fig. 8). The fluorescence signal of FNA-Fe$_3$O$_4$ in response to different concentrations of SNCA mRNA increased monotonically from 0 to 30 nM (Supplementary Fig. 9). Although the fluorescence signal was partially quenched in the presence of AA and H$_2$O$_2$, it still maintained a good response to SNCA mRNA (Supplementary Fig. 10). Other control mRNAs, including β-synuclein (SNCB) mRNA, γ-synuclein (SNCG) mRNA, and mouse SNCA mRNA, led to lower fluorescent signal changes of FNA-Fe$_3$O$_4$ (Supplementary Fig. 11). These results suggested the potential of FNA-Fe$_3$O$_4$ for precise intracellular SNCA mRNA imaging.

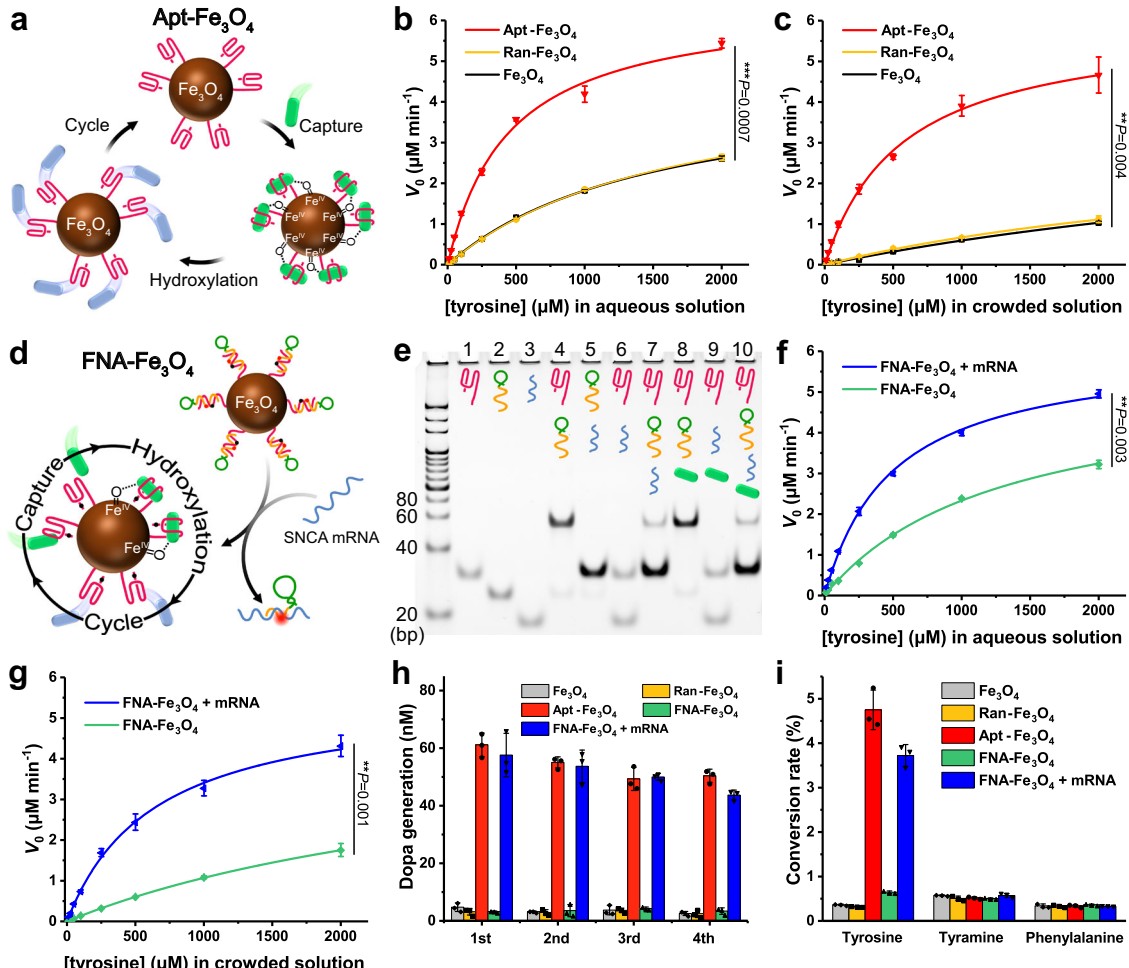

**Fig. 3 | Characterizations of artificial enzymes and their catalytic and fluorescence imaging performances. a** Schematic diagram of Apt-Fe₃O₄ catalysis. **b, c** Rates of hydroxylation of tyrosine to Dopa by Fe₃O₄, Apt-Fe₃O₄ and Ran-Fe₃O₄ in the presence of H₂O₂, AA and variable concentrations of tyrosine in aqueous solution (**b**) and in crowded solution (**c**). **d** Schematic diagram of strand displacement responsiveness and catalysis of FNA-Fe₃O₄. **e** PAGE analysis of the functional nucleic acids structure and the strand displacement reaction (red strand: tyrosine aptamer; green-yellow strand: block strand; blue strand: SNCA mRNA; green substrate: tyrosine). Gel was representative of $n$ = 3 independent experiments. **f, g** Rates of hydroxylation of tyrosine to Dopa by FNA-Fe₃O₄ with or without mRNA in the presence of H₂O₂, AA and variable concentrations of tyrosine in aqueous solution (**f**) and in crowded solution (**g**). **h** The typical cyclic process of tyrosine capture and hydroxylation by 10 μg mL⁻¹ artificial enzymes in the presence of 100 μM tyrosine, 5 mM H₂O₂ and 5 mM AA. **i** Catalytic selectivity of 10 μg mL⁻¹ artificial enzymes for hydroxylation of 100 μM tyrosine and interference molecules in the presence of 5 mM H₂O₂ and 5 mM AA. **b, c, f, g, h** and **i** The results were expressed as mean ± SD ($n$ = 3 independent experiments). **b, c, f** and **g** $P$-values were calculated by two-tailed $t$-test. Source data are provided as a Source Data file.

The tyrosine hydroxylase-mimicking activity of FNA-Fe₃O₄ in the presence or absence of SNCA mRNA was also subsequently measured (Supplementary Figs. 12, 13). As shown by the kinetic curve in Fig. 3f, g, FNA-Fe₃O₄ exhibited high catalytic activity after strand displacement reaction with SNCA mRNA in aqueous and crowded solutions (kinetic parameters were listed in Supplementary Table 2). And the catalytic activity of FNA-Fe₃O₄ alone stayed at a relatively low level, especially in crowded solution. This feature provided the potential that FNA-Fe₃O₄ would not affect dopamine homeostasis in normal cells, and only supplied Dopa in cells with abnormal overexpression of SNCA mRNA.

After reacting with SNCA mRNA, FNA-Fe₃O₄ could successively capture tyrosine in tyrosine solution and catalyze tyrosine hydroxylation in solution containing H₂O₂ and AA, and the Dopa generation within 5 min of reaction remained stable during four repetitions (Fig. 3h), which would be substantial proof of the typical cyclic catalytic process of the artificial enzymes. In addition, several common intracellular small molecules (tyramine and phenylalanine) capable of aromatic ring hydroxylation were used as model interferences[45], to test the substrate selectivity of artificial enzymes to tyrosine. As shown in Fig. 3i, FNA-Fe₃O₄ + mRNA exhibited much higher hydroxylation

conversion rates for tyrosine than the other interfering substances within 5 min of reaction, showing excellent selectivity for tyrosine and proving their intracellular anti-interference ability. Moreover, no DNA fragments were generated in the solutions of Apt-Fe₃O₄, FNA-Fe₃O₄, or FNA-Fe₃O₄ + mRNA (10 μg mL⁻¹) containing tyrosine, H₂O₂, and AA after 30 min of reaction, demonstrating the stability of DNA strands during the reaction (Supplementary Fig. 14).

## Intracellular biological effects of FNA-Fe₃O₄
BBB is the first obstacle for FNA-Fe₃O₄ to enter dopaminergic neurons. Therefore, an in vitro BBB model was established with TfR-expressing immortalized mouse cerebral endothelial cells (bEnd.3)[46], and the artificial enzymes were added in the apical side to investigate the penetration efficiency (Fig. 4a). FNA-Fe₃O₄ gradually accumulated in the basolateral side over time, and the transport efficiency rose to 21.8% after 120 min (Fig. 4b). In contrast, Apt-Fe₃O₄, Ran-Fe₃O₄, and Fe₃O₄ without TfR aptamers accumulated slowly in the basolateral side. These results suggested that FNA-Fe₃O₄ could efficiently cross the in vitro BBB model, in which the TfR aptamer played an important role. To explore the influence of blood circulation process on the BBB

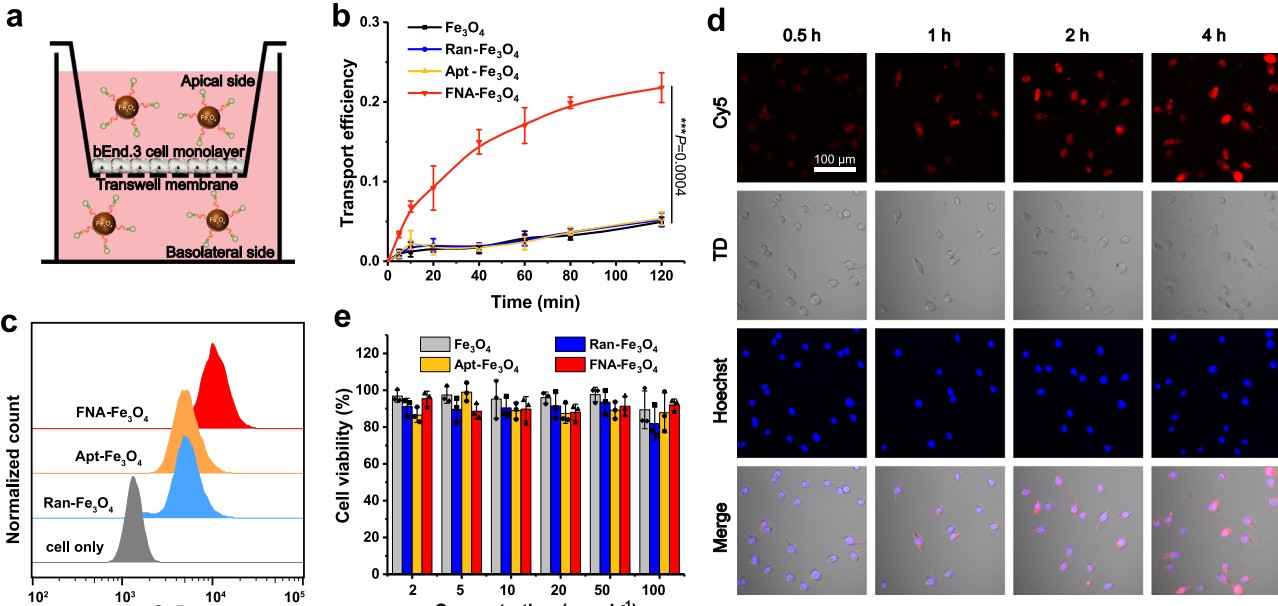

**Fig. 4 | Cellular availability of the artificial enzymes. a** Schematic of FNA-Fe₃O₄ crossing the in vitro BBB model. **b** Transport efficiency of the artificial enzymes across the BBB in vitro. **c** Flow cytometry analysis of cellular uptake of the artificial enzymes. **d** Confocal fluorescence images of the gradual cellular internalization of FNA-Fe₃O₄ at different time interval. **e**, Cell viability of BV2 cells incubated with the artificial enzymes at different concentrations. **b**, **e** The results were expressed as mean ± SD ($n = 3$ biologically independent samples). **b** $P$-values were calculated by two-tailed $t$-test. **c**, **d** One representative data was shown from three independently repeated experiments. Source data are provided as a Source Data file.

crossing of FNA-Fe₃O₄, the penetration efficiency was further measured after FNA-Fe₃O₄ was incubated with mouse blood for 4 h. As shown in Supplementary Fig. 15, FNA-Fe₃O₄ still retained its ability to penetrate the in vitro BBB model after whole blood co-incubation, which proved the potential of FNA-Fe₃O₄ for further brain application.

After verifying the BBB crossing of FNA-Fe₃O₄, we further examined its internalization in neurons. It is reported that the diseased dopaminergic neurons in PD also express TfR[47]. Therefore, we used TfR-positive mouse microglia BV2 cells as a cellular model in PD[48]. Cy5-modified fluorescence-on FNA-Fe₃O₄ without BHQ2 was employed, and fluorescence-on Apt-Fe₃O₄ and Ran-Fe₃O₄ as controls (Supplementary Fig. 16). In comparison with Apt-Fe₃O₄ and Ran-Fe₃O₄ without TfR aptamer, FNA-Fe₃O₄ showed a better binding shift toward the target BV2 cells (Fig. 4c and Supplementary Fig. 17). The strong Cy5 fluorescence signal representing FNA-Fe₃O₄ was localized predominantly in the cytoplasm of BV2 cells, and remained at high intracellular level for a longer time (Fig. 4d). The cell viability test results illustrated that FNA-Fe₃O₄ hardly caused observable cytotoxicity toward BV2 cells (Fig. 4e), suggesting good biocompatibility with brain endothelial cells. These properties provided the basis for the intracellular application of FNA-Fe₃O₄.

To assess the function of nucleic acid components on FNA-Fe₃O₄, we designed two control nanomaterials. Cat-Fe₃O₄ represented only the effect of tyrosine hydroxylase-mimicking activity and ASO-Fe₃O₄ represented antisense oligonucleotide therapy, respectively (Fig. 5a, b and Supplementary Fig. 18). FNA-Fe₃O₄ and these control nanomaterials maintained good biocompatibility toward the neuron model human neuroblastoma SH-SY5Y cells (Supplementary Fig. 19). Next, we validated the tyrosine hydroxylase-mimicking properties of the artificial enzymes in cell models. Abnormal overexpression of the SNCA gene is a feature of PD cells[34]. Thus, SH-SY5Y cells transfected with SNCA-EGFP gene were regarded as PD dopaminergic neurons, which overexpressed SNCA gene and expressed enhanced green fluorescent protein (EGFP) as a marker. Original SH-SY5Y cells were regarded as normal neurons for control. Dopa has an extremely short half-life due to intracellular active Dopa decarboxylase activity, thus it is usually monitored by the content of its decarboxylation product, dopamine[49]. As shown in Fig. 5c, FNA-Fe₃O₄ treated normal cells displayed little change in intracellular dopamine content, indicating that it would not interfere with dopamine homeostasis in normal neurons. In the FNA-Fe₃O₄ treated PD cells, the dopamine content illustrated a gradual upward trend within 8 h, and then reached a stable plateau. This outcome demonstrated that FNA-Fe₃O₄ could exert a tyrosine hydroxylase-mimicking function in cells in response to abnormally overexpression of SNCA mRNA for cell-specific spatial control. Meanwhile, Cat-Fe₃O₄ also showed similar tyrosine hydroxylase-mimicking activity, while ASO-Fe₃O₄ did not, which reflected the decisive role of tyrosine aptamers on the catalysis of the artificial enzyme. We also tried the effect of direct Dopa incubation (0.5, 2, 8 μM) on intracellular dopamine levels. After 2 h incubation and and replacing the culture medium to simulate the short plasma half-life of dopamine, the intracellular dopamine content gradually decreased (Fig. 5d). Higher Dopa doses maintained higher dopamine levels for longer periods, but also resulted in more dramatic fluctuations in dopamine contents. In general, longer-lasting enhanced dopamine levels implied a more prolonged effect of Dopa treatment, whereas dramatic fluctuations in dopamine levels herald disrupted dopamine homeostasis in neurons[10]. This was the root cause of the inability to balance the therapeutic effects and side effect controls of Dopa. In contrast, the in situ continuous Dopa production provided by FNA-Fe₃O₄ expressed longer duration and more acceptable dopamine fluctuations. It should be mentioned that since the effective physiological concentrations of Dopa and dopamine were much lower than those of tyrosine, the application of FNA-Fe₃O₄ for tyrosine hydroxylation would not affect the intracellular tyrosine homeostasis (Supplementary Fig. 20).

Hybridization of the block strand to SNCA mRNA also reduces the SNCA expression. After 36 h of artificial enzymes co-incubation, FNA-Fe₃O₄ and ASO-Fe₃O₄ reduced the expression levels of SNCA mRNA (Supplementary Fig. 21) and protein (Fig. 5e) in PD cell model, which was beneficial for delaying neuronal degeneration. Meanwhile, the strand displacement reaction on FNA-Fe₃O₄ induced by SNCA mRNA provides an opportunity for intracellular fluorescence imaging of

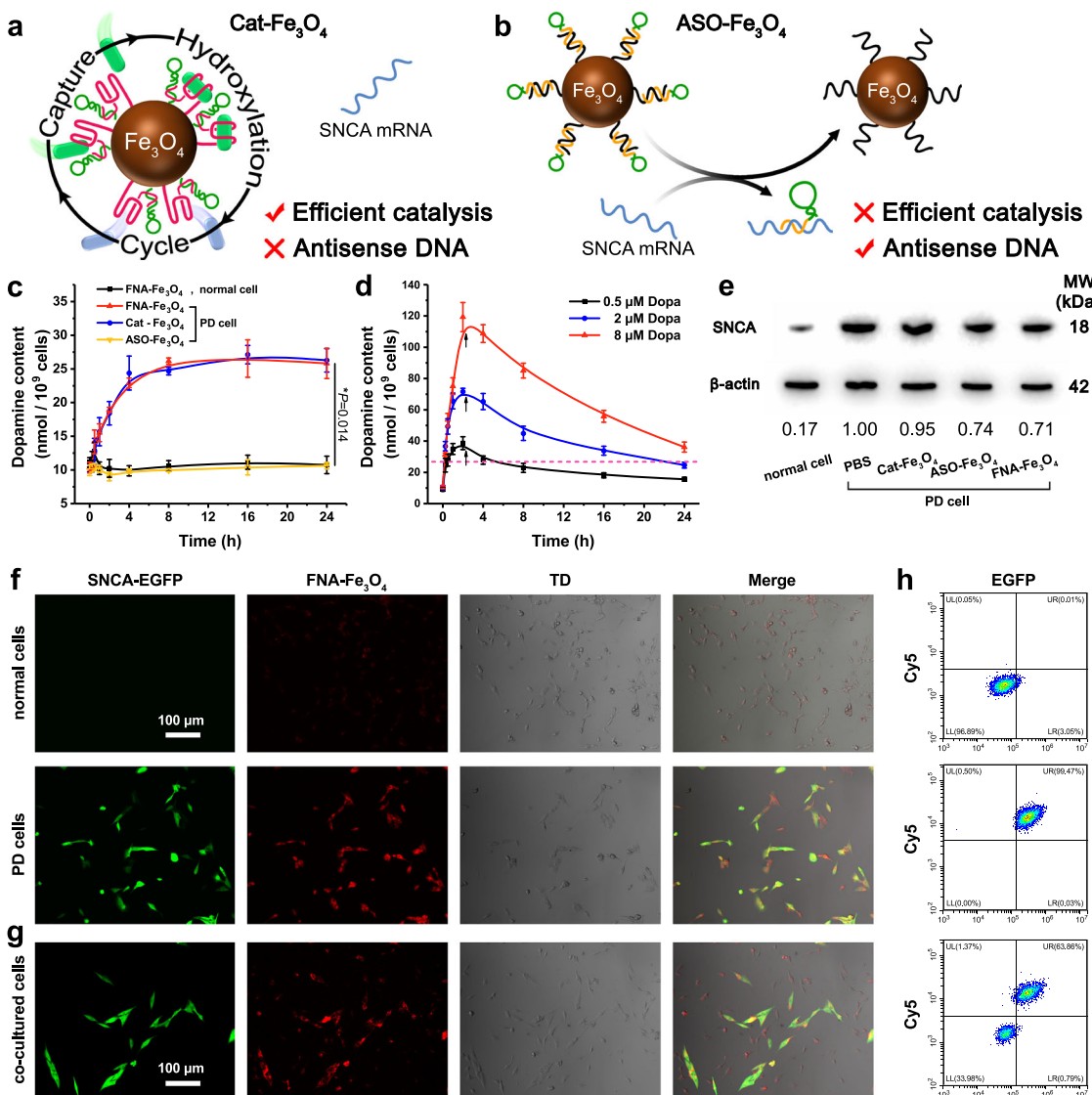

**Fig. 5 | Intracellular biological effects of the artificial enzymes. a** Schematic diagram of Cat-Fe$_3$O$_4$ Catalysis. **b** Schematic diagram of strand displacement of ASO-Fe$_3$O$_4$. **c** Artificial enzymes mediated changes in intracellular dopamine levels. **d** Dopa incubation induced dramatic fluctuations in intracellular dopamine levels. Arrows represent Dopa removal time point (adding fresh culture medium). **e** SNCA expression in different treated SH-SY5Y cells. **f, g** Confocal imaging of normal cells (without EGFP fluorescence) and PD cells (express EGFP fluorescence) (**f**) and co-cultured cells by FNA-Fe$_3$O$_4$ (**g**). **h** Flow cytometry analysis corresponding to each panel in (**f**) and (**g**). **c, d** The results were expressed as mean ± SD ($n$ = 3 biologically independent samples). **c** P-values were calculated by two-tailed $t$-test. **e-h** one representative data was shown from three independently repeated experiments. Source data are provided as a Source Data file.

SNCA mRNA. In the fluorescence analysis of Fig. 5f, the cells exhibiting the green fluorescence signal of EGFP were SNCA mRNA-overexpressing cells, which could be displayed by FNA-Fe$_3$O$_4$ in Cy5 red fluorescence (Supplementary Fig. 22). FNA-Fe$_3$O$_4$ showed different fluorescence signal intensities for co-cultured normal cells and PD cells, demonstrating that FNA-Fe$_3$O$_4$ could differentiate normal and PD neurons by fluorescence imaging in response to overexpressed SNCA mRNA (Fig. 5g). The consistent fluorescence analysis results were obtained from corresponding flow cytometry (Fig. 5h and Supplementary Fig. 23). These results suggested that FNA-Fe$_3$O$_4$ was able to monitor the intracellular SNCA mRNA expression levels.

### In vivo biological effects of FNA-Fe$_3$O$_4$

Encouraged by the superior performance of FNA-Fe$_3$O$_4$ in cell experiments, its blood circulation and biodistribution were further evaluated. After FNA-Fe$_3$O$_4$ was incubated in blood for 4 h, the displaced block strand remained stable characterized by the mass spectra (Supplementary Fig. 24), which enabled FNA-Fe$_3$O$_4$ to maintain its functional nucleic acid structure during blood circulation for brain targeting. After wild C57BL/6 N mice received tail vein injection of fluorescence-on FNA-Fe$_3$O$_4$, the fluorescence signal from FNA-Fe$_3$O$_4$ was observed to accumulate rapidly in the brain within 4 h (Supplementary Fig. 25). The blood circulation profile of FNA-Fe$_3$O$_4$ was shown in Supplementary Fig. 26, and the half-life of blood circulation was about 1.2 h. Ex vivo organ fluorescence images showed that FNA-Fe$_3$O$_4$ also accumulated in a large amount in the liver, and in a small amount in the kidney, lung and spleen (Supplementary Fig. 27). Furthermore, immuno-fluorescence images of substantia nigra sections of brain revealed that FNA-Fe$_3$O$_4$ could be delivered to tyrosine hydroxylase-positive neuron region (Supplementary Fig. 28), which enabled it to effectively participate in dopaminergic pathway. These results indicated acceptable brain utilization of FNA-Fe$_3$O$_4$.

FNA-Fe$_3$O$_4$ exhibited concentration dependent fluorescence signal response to SNCA mRNA without obvious attenuation after 4 h of in vitro blood incubation (Supplementary Fig. 29), thus it can be used for in situ fluorescence imaging of overexpressed SNCA mRNA of PD

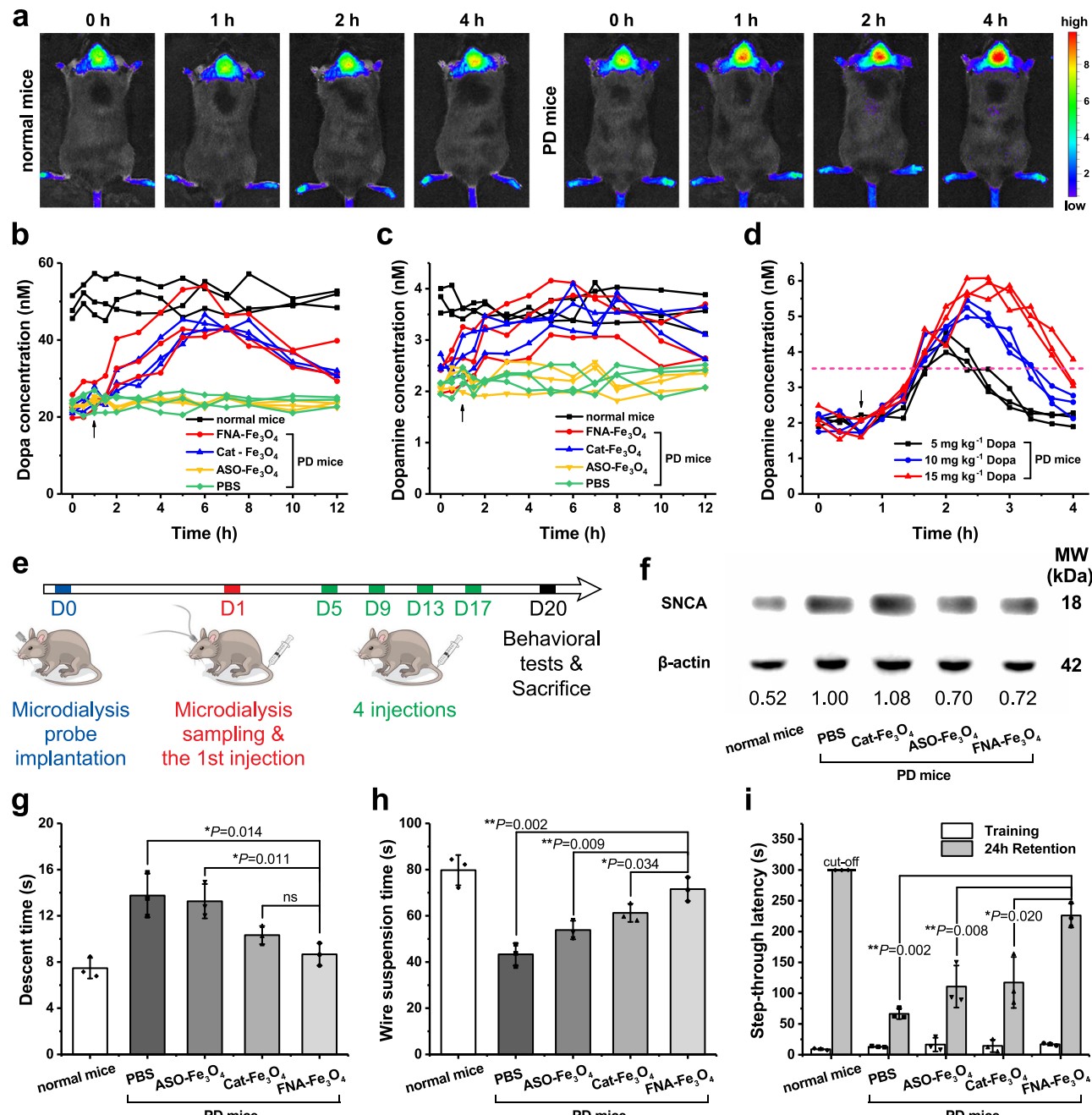

**Fig. 6 | In vivo biological effects of the artificial enzymes. a** In vivo fluorescence imaging of FNA-Fe₃O₄ in normal mice and PD mice. **b**, **c** Artificial enzymes mediated changes in Dopa levels (**b**) and dopamine levels (**c**) in mice striatal microdialysates (arrows represent dosing time point, each line represent one mouse). **d** Dopa administration induced dramatic fluctuations in dopamine levels in mice striatal microdialysates (dash line represents the mean dopamine level of normal mice in (**c**), each line represent one mouse). **e** Schematic timeline of microdialysis, injections and behavioral tests. **f** The expression of SNCA in the brain tissue of differently treated mice. **g** Descent time spent by the mice in the pole test. **h** Wire suspension time of the mice in the wire suspension test. **i** Step-through latency time of the mice in the step-through passive avoidance test. **a**, **f** One representative data was shown from three independently repeated experiments. **b–d** each line represent one mouse, $n = 3$ biologically independent animals. **g–i** The results were expressed as mean ± SD ($n = 3$ biologically independent animals), $P$-values were calculated by two-tailed $t$-test. Source data are provided as a Source Data file.

in vivo. Wild mice, and Thy1-SNCA transgenic mice which carries human wild type SNCA driven by the murine thymus cell antigen 1 (Thy1) promoter, were used as normal and PD animal models, respectively[35]. Compared with normal mice, gradually enhanced Cy5 fluorescence signal was observed in the PD mice brain within 4 h after tail vein injection of FNA-Fe₃O₄ (Fig. 6a), pointing that FNA-Fe₃O₄ could realize imaging of PD animal model.

Subsequently, striatal microdialysis was performed during tail vein injection of Cat-Fe₃O₄, ASO-Fe₃O₄, or FNA-Fe₃O₄ (4 mg kg⁻¹) in PD

mice to analyze artificial enzyme-catalyzed in situ Dopa production. Cat-Fe₃O₄ and FNA-Fe₃O₄, which had tyrosine hydroxylase-mimicking properties, elevated Dopa and dopamine levels in brain tissue to levels slightly below normal, whereas ASO-Fe₃O₄ did not induce changes in Dopa and dopamine levels (Fig. 6b, c). This trend suggested that the artificial enzyme mediated the restoration of dopamine series neurotransmitter production in the brain. Correspondingly, microdialysis results during Dopa administration were compared. The uptrend of dopamine was only sustained for a short period of time (within about

4 h) and fluctuated widely (Fig. 6d), thus showing the superiority of the artificial enzyme strategy. Moreover, FNA-Fe$_3$O$_4$ did not affect the tyrosine homeostasis in brain (Supplementary Fig. 30). After a 20 days treatment period (Fig. 6e), ASO-Fe$_3$O$_4$ and FNA-Fe$_3$O$_4$ with antisense oligonucleotide moieties considerably reduced SNCA expression in brain tissues, while the influence of Cat-Fe$_3$O$_4$ on overexpressed SNCA was negligible (Fig. 6f). This result revealed the importance of the antisense oligonucleotide moiety on the artificial enzyme for the regulation of overexpressed SNCA.

Continuous in situ Dopa production and mitigation of aberrant SNCA overexpression together work on PD pathology features in Thy1-SNCA mice. Therefore, the motor coordination ability of the mice was further evaluated by the pole climbing test and the wire suspension test (Fig. 6g, h). It can be seen that the motor coordination ability of PD mice was significantly worse than that of healthy mice. Cat-Fe$_3$O$_4$ and ASO-Fe$_3$O$_4$ partially restored motor performance in PD mice, reflecting the alleviation of PD symptoms by tyrosine hydroxylase replacement therapy and antisense oligonucleotide therapy, respectively. The therapeutic effect of Cat-Fe$_3$O$_4$ alone was slightly better than that of ASO-Fe$_3$O$_4$, because in situ dopa supply can directly act on dopaminergic nervous system. The contribution of antisense oligonucleotide therapy was to alleviate the PD pathological process, thus FNA-Fe$_3$O$_4$ treated mice exhibited better performance than monotherapy. In addition, PD mice can exhibit a decline in learning and memory, which could be tested by passive avoidance test. Longer step-through latency in 24 h retention represented more complete learning and memory. Administration of Cat-Fe$_3$O$_4$ and ASO-Fe$_3$O$_4$ in PD mice helped to ameliorate the decline in learning and memory at a similar level, and FNA-Fe$_3$O$_4$ treatment was more pronounced (Fig. 6i, Supplementary Fig. 31 and Supplementary Video 1). These results demonstrated that FNA-Fe$_3$O$_4$ could promote the recovery of dyskinesia and memory deficits in PD mice model, and the therapeutic effect was derived from the synergy of tyrosine hydroxylase-mimicking activity and antisense oligonucleotide.

After 20 days treatment period, the peripheral tissues of mice exhibited Dopa and dopamine homeostasis levels similar to those of the untreated group (Supplementary Figs. 32, 33). Hematoxylin-eosin (H&E) stained sections of each organ showed no tissue lesions (Supplementary Fig. 34). The stable levels of liver function markers (alanine transaminase (ALT), aspartate aminotransferase (AST)) and kidney function marker (blood urea nitrogen (BUN)) in serum indicated no visible hepatic or renal toxicity during treatment (Supplementary Fig. 35). Negligible hemolytic activity of FNA-Fe$_3$O$_4$ when incubated with red blood cells was shown, which indicated good hemocompatibility (Supplementary Fig. 36). And no significant fluctuations in mouse body weight were found during the treatment (Supplementary Fig. 37). These results proved that the presented artificial enzyme own favorable biocompatibility with minimal peripheral tissue side effects.

## Discussion
Oral Dopa treatment is currently the most effective mean for clinical improvement in PD symptoms, yet the modality of Dopa therapy remains to be optimized due to Dopa-induced dyskinesia[5–10]. Starting from the pathological features of PD, the replacement of inactivated tyrosine hydroxylase by artificial enzyme is expected to provide smooth Dopa production.

Iron-based catalysts have been reported to catalyze the hydroxylation of tyrosine and achieve high catalytic conversion[24,25]. Based on the further investigation of nanozymes, we proved that Fe$_3$O$_4$ could generate Fe(IV)=O active intermediates. This active intermediate is consistent with the active intermediate in tyrosine hydroxylase, which can hydroxylate tyrosine to produce Dopa. However, in the intracellular molecular crowding environment, Fe$_3$O$_4$ is difficult to exhibit the specific and efficient catalytic ability, which limits its application[26]. Therefore, we set out to combine Fe$_3$O$_4$

with tyrosine aptamers to construct tyrosine aptamer-based artificial enzyme Apt-Fe$_3$O$_4$, which can replace tyrosine hydroxylase and enable the continuous and steady conversion of tyrosine to Dopa in cells.

To promote neuronal enrichment and avoid the side effects of Dopa production on peripheral tissues, the functional nucleic acid-based responsive artificial enzyme FNA-Fe$_3$O$_4$ was further designed. FNA-Fe$_3$O$_4$ was verified to potently penetrate the BBB and was enriched in neurons by virtue of the TfR aptamer. Only in PD neurons, FNA-Fe$_3$O$_4$ exhibited efficient tyrosine hydroxylase-mimicking activity, supplemented by antisense oligonucleotide and fluorescence imaging. FNA-Fe$_3$O$_4$ mediated continuous and steady Dopa production and the decrease of pathological SNCA synergistically alleviated the pathological symptoms in cell and mice PD models, which was finally reflected by obviously improved animal behavior test results. In contrast to the conventional Dopa administration strategy for PD, the presented responsive artificial enzyme FNA-Fe$_3$O$_4$ not only effectively ameliorates motor symptoms and memory deficits in PD model mice, but also avoids dramatic fluctuation of Dopa concentration, providing an outstanding paradigm for the advanced therapeutic strategy of PD.

## Methods
### Ethical statement

All animal experiments were approved by the Institutional Animal Care and Use Committee of Fuzhou University (Approval ID: 2022-SG-014). All research was performed in accordance with relevant guidelines and regulations.

### Materials
15 nm spherical Fe$_3$O$_4$-COOH water dispersed nanoparticles were purchased from Ocean NanoTech LLC. All nucleic acids sequences used in the study were commercially ordered (Sangon Biotech (Shanghai) Co., Ltd.). The nucleic acid sequences used in this study are listed in Supplementary Table 1.

### Preparation of Apt-Fe$_3$O$_4$ and Ran-Fe$_3$O$_4$
A 40 μL solution of Fe$_3$O$_4$-COOH (5 mg mL$^{-1}$) were added to a 100 μL MES buffer (5 mM, pH 5.5) containing EDC (10 mM) and sulfo-NHS (25 mM) and left to react for 15 min. The above mixture was subsequently added 100 μL PBS (100 mM, pH 7.2) and 40 μL of 5'-terminal amino-modified tyrosine aptamer or control strands (100 μM in 100 mM PBS, pH 7.2). The amide coupling reaction was performed at room temperature for 12 h. The precipitates after the reaction were purified by magnetic separation, and washed 4 times.

### Preparation of FNA-Fe$_3$O$_4$
A 40 μL solution of Apt-Fe$_3$O$_4$ (5 mg mL$^{-1}$) were added to 400 μL of block strand solution (10 μM in 50 mM phosphate buffer, pH 7.2, containing 100 mM NaCl and 5 mM MgCl$_2$), and kept at 37 °C for 3 h. The precipitates after the reaction were purified by magnetic separation and washed 4 times.

### Measurements of Dopa and tyrosine concentrations
Dopa production and tyrosine consumption in the reaction solutions were determined by UPLC-MS (Acquity UPLC H-Class, Waters; Xevo G2-XS Qtof, Waters). First, sample solutions containing Fe$_3$O$_4$ (10 μg mL$^{-1}$), tyrosine (100 μM), AA (5 mM) and H$_2$O$_2$ (5 mM) were prepared. After reacting for different times, sample solutions were filtered and diluted appropriately, and injected into a C18 chromatographic column set at 35 °C. MS-grade acetonitrile/water (10%/90%) was used as the mobile phase and run at a flow rate of 0.3 mL min$^{-1}$. The peak areas of m/z = 198.08 and 182.08 in the positive ion mode were used as the data for Dopa and tyrosine quantification, respectively.

## ESR measurements

Radical species such as ·OH and ·OOH, were detected using the ESR spin trapping technique coupled with a spin trap 5,5-dimethyl-1-pyrroline N-oxide (DMPO). Typically, five mixtures: $Fe_3O_4$ (10 µg mL$^{-1}$) and $H_2O_2$ (5 mM); $Fe_3O_4$ (10 µg mL$^{-1}$) and AA (5 mM); $Fe_3O_4$ (10 µg mL$^{-1}$) and $H_2O_2$ (5 mM) and AA (5 mM); $Fe_3O_4$ (10 µg mL$^{-1}$) and $H_2O_2$ (5 mM) and AA (5 mM) and tyrosine (100 µM); $H_2O_2$ (5 mM) and AA (5 mM), were prepared in phosphate buffer (pH 7.2, containing 100 mM NaCl and 5 mM $MgCl_2$), including DMPO (0.88 M). The ESR measurements were taken with the following conditions: Microwave Power, 6.38 mW; Modulation Amplitude, 1.00 G; Modulation Frequency, 100.00 kHz; Conversion Time, 41.00 ms.

## Qualitative detection of Fe(IV)=O with DMSO as probe

MSM production in the reaction solutions were determined by GC-MS (QP-2020NX, Shimadzu). Sample solutions containing $Fe_3O_4$ (10 µg mL$^{-1}$), DMSO (1 mM), AA (5 mM) and $H_2O_2$ (5 mM) were prepared. After reacting for different times, $Fe_3O_4$ was magnetically separated and 0.3 µL sample solution was injected into HP-INNOWax chromatographic column set at 250 °C. The peak of m/z = 79.00 was used as the qualitative data of MSM.

## Evaluation of the loading of the tyrosine aptamers on Apt-Fe$_3$O$_4$

The ratios between the Apt-Fe$_3$O$_4$ and tyrosine aptamer could be determined by the fluorescent label. 25 µg of Apt-Fe$_3$O$_4$ was hybridized with 0.5 nmol of Cy5-labeled tyrosine aptamer complementary strand (sequence shown in Supplementary Table 1) in 1 ml PBS (pH 7.2) at 37 °C for 3 h. The reaction solution was then magnetic separated, and the Cy5 fluorescence spectrum of the supernatant was measured. The calibration curve of the Cy5-labeled tyrosine aptamer complementary strand was determined by the fluorescence intensity at 665 nm, and the amount of tyrosine aptamer loaded on Apt-Fe$_3$O$_4$ was calculated. It should be noted that multiple groups executed in the experiments were achieved similar ratio of aptamers to Fe$_3$O$_4$.

## Kinetic measurements with artificial enzymes

For the tyrosine hydroxylation process, the artificial enzymes (final concentration 10 µg mL$^{-1}$), AA (5 mM) and $H_2O_2$ (5 mM) were dispersed in 50 mM phosphate buffer (pH 7.2, containing 100 mM NaCl and 5 mM $MgCl_2$). And variable concentrations of tyrosine (10-2000 µM) were added to the mixture. In the simulation of macromolecular crowding, additional 20 wt% PEG-20000 was added to the reaction solution. After the start of the reaction, the Dopa concentrations in the reaction mixture were monitored by HPLC with fluorescence detector (ex 277 nm, em 310 nm) at each time point. The samples were injected into a C18 chromatographic column with acetonitrile/water (2%/98%) at a flow rate of 1 mL min$^{-1}$ as the mobile phase. Catalytic reaction kinetics were fitted by tyrosine concentrations and HPLC measured Dopa generation rates.

## Fluorescence detection of strand displacement reaction on artificial enzymes

10 µg mL$^{-1}$ FNA-Fe$_3$O$_4$ was added with different concentrations (0−30 nM) of SNCA mRNA, and incubated at 37 °C for 2 h. Fluorescence spectra were then collected under excitation at 620 nm wavelength. The analogous mRNAs used are listed in Supplementary Table 1.

## Cell culture

SH-SY5Y and bEnd.3 cell lines were purchased from American Type Culture Collection (cat# CRL-2266 for SH-SY5Y, cat# CRL-2299 for bEnd.3). BV-2 cell line and SNCA-EGFP plasmid were purchased from Hunan Fenghui Biotechnology Co., Ltd (cat# CL0056 for BV2). SH-SY5Y and SNCA-EGFP gene-transfected SH-SY5Y cells were cultured in DMEM-F12 with 15% fetal bovine serum (FBS, Gibco) and 1% penicillin-streptomycin (PS, Meilunbio). BV2 and bEnd.3 cells were grown in DMEM supplemented with 10% FBS and 1% PS. Both cell lines were incubated at 37 °C in 5% $CO_2$.

## Transport study in the constructed in vitro BBB cell model

To study the transport efficiency of the artificial enzymes across the in vitro BBB cell model, a cell monolayer system was established. Transwell membrane inserts were first placed in the 12-wells culture plates and moistened with 1.2 mL culture medium in the basolateral side. Then bEnd.3 cells ($2 \times 10^5$) were seeded in the apical side of transwell inserts and cultured at 37 °C in 5% $CO_2$. The fresh culture medium was replaced every other day, until cells form a very compact monolayer, a BBB cell model. After the last replacement of fresh culture medium, each artificial enzymes (100 µg mL$^{-1}$) were added to the apical side and co-incubated with the cell monolayer for 5, 10, 20, 40, 60, 80 and 120 min. At each time point, culture medium were taken from the basolateral side and the signal of Fe was detected by ICP-MS. Since the Fe content was in direct proportion to the amount of artificial enzymes, the transport efficiency of the artificial enzymes across the BBB cell model was calculated by the ratio of the basolateral Fe signal to the total Fe signal of the 100 µg mL$^{-1}$ artificial enzyme. For the stability exploration in blood circulation process, FNA-Fe$_3$O$_4$ was incubated with anticoagulated mice whole blood for 4 h at room temperature. Then FNA-Fe$_3$O$_4$ was magnetically separated and the transport efficiency was further measured.

## Cellular internalization of the artificial enzymes

In the flow cytometry internalization test, $2 \times 10^5$ BV2 cells were seeded into 6-well plates for 24 h. Then, cells were incubated with each fluorescence-on artificial enzyme samples (100 µg mL$^{-1}$ in DMEM medium) for 1 h. Afterwards, the cells were washed with iced 1× PBS three times and suspended in 100 µL 1× PBS buffer for fluorescence analysis on flow cytometer (Beckman Coulter Cytoflex). The flowcytometry data were analyzed on Flowjo software (version 10.0.7).

In the confocal microscopy study, $2 \times 10^5$ BV2 cells were seeded into 20 mm glass bottom cell culture dishes for 24 h. Then, cells were incubated with fluorescence-on FNA-Fe$_3$O$_4$ (100 µg mL$^{-1}$ in DMEM medium) for 0.5, 1, 2 and 4 h. Afterwards, the cells were incubated with hoechst 33342 for cell nucleus staining, followed by washing three times with iced 1× PBS. Fluorescence images were captured by the confocal laser scanning microscope (CLSM, Nikon A1 + SIM Ti2-E).

## Cell viability study

Cell viability of the artificial enzymes on BV2 cells was determined by Cell Counting Kit-8 (CCK-8). Cells were seeded into 96-well plates (8000 cells per well) for 24 h, then the cells were incubated with each artificial enzyme sample (100 µg mL$^{-1}$ in 100 µL medium) for 8 h. Then, the medium was removed from each well, followed by washing with 1× PBS. The cells were continued to be cultured in fresh medium (100 µL) at 37 °C for 6 h, and a CCK-8 working solution was used to replace the culture medium and cultured for another 15 min. Finally, a microplate reader was used to measure the absorbance at 450 nm of each well. Cell viability was calculated by comparing with the absorbance of the untreated cells and expressed as the percentage histogram.

## Measurements of intracellular dopamine and tyrosine content

$2 \times 10^5$ SH-SY5Y or SNCA-EGFP gene-transfected SH-SY5Y cells were seeded into 6-well plates and cultured in DMEM-F12 medium containing 0.5 mM AA for 24 h. Then, the cells were incubated with each artificial enzyme sample (100 µg mL$^{-1}$ in 1 mL medium) for 0.25, 0.5, 1, 2, 4, 8, 16 and finally 24 h. For cells incubated with Dopa (0.5, 2, 8 µM in 1 mL medium), the cells were incubated in the Dopa containing medium for the first 2 h, and then the medium was replaced with fresh medium without Dopa. At each time point, the cells were collected and counted, and then resuspended in 200 µL 0.1 M $HClO_4$ containing

10 mM sodium disulfite and 1 mM EDTA. After sonication, lysates were centrifuged at 17,000 g for 30 min. The supernatants were analyzed by HPLC with fluorescence detector (ex 277 nm, em 310 nm). The samples were injected into a C18 chromatographic column with acetonitrile/water (2%/98%) at a flow rate of 1 mL min$^{-1}$ as the mobile phase.

## Quantitative RT-PCR of total RNA extracted from SH-SY5Y cells

Total RNA was extracted from SH-SY5Y cells using RNA Extraction (Servicebio). RNA was reverse transcribed to complementary DNA (cDNA) using the SweScript RT I First Strand cDNA Synthesis Kit (Servicebio) according to the manufacturer's protocol and cDNA was used as the template for qPCR. qPCR was performed on CFX Connect real time PCR system (Bio-Rad). Relative mRNA expression was normalized by the GAPDH RNA level in each sample and calculated using the delta-delta Ct method.

## Immunoblot assay of SNCA

$10^5$ SH-SY5Y or SNCA-EGFP gene-transfected SH-SY5Y cells were seeded into 6-well plates for 24 h. Then, the cells were incubated with each artificial enzyme sample (100 μg mL$^{-1}$ in 1 mL medium) for 36 h. Then, cells were lysed in cell lysis buffer for Western containing protease inhibitor cocktail set III (beyotime). The protein concentrations in lysates were quantified by BCA protein assay kit. 15 μg protein extracts were separated using 10% SDS-polyacrylamide gels. The proteins in the gel after electrophoresis were transferred onto PVDF membrane, and the nonspecific binding sites were blocked by 5% milk/TBST for 1.5 h. The membranes were incubated with the primary antibody for SNCA (Cell Signaling Technology, validated by manufacturer, cat# 51510, clone number E4U2F, 1:1000 dilution) at 4 °C overnight, washed several times, and then incubated with the HRP-labeled anti-rabbit secondary antibody (Cell Signaling Technology, validated by manufacturer, cat# 7074, 1:1000 dilution) for 1 h, and then subjected to ECL detection analysis.

To quantify SNCA in mouse brains, total tissue proteins were extracted from the cerebral hemispheres using the above-mentioned cell lysis buffer. The subsequent operation methods were performed as described above.

## Fluorescence analysis of cellular SNCA mRNA by artificial enzymes

In the confocal microscopy fluorescence imaging study, $2 \times 10^5$ SH-SY5Y, SNCA-EGFP gene-transfected SH-SY5Y, or a one-to-one mix of cells were seeded into 20 mm glass bottom cell culture dishes for 24 h. Then, cells were incubated with FNA-Fe$_3$O$_4$ (100 μg mL$^{-1}$ in DMEM-F12 medium) for 2 h. Afterwards, the cells were washed three times with 1× PBS. Fluorescence images were captured by the CLSM in EGFP and Cy5 channel.

In the flow cytometry fluorescence imaging test, the corresponding cells were seeded into 6-well plates. Following the same artificial enzyme incubation and washing process, cells were collected and resuspended in 1× PBS buffer for fluorescence analysis. The flow-cytometry data were analyzed on CytExpert software (version 2.4.0.28, Beckman Coulter Inc.).

## Mouse strains

Wild C57BL/6 N (6 months old), and Thy1-SNCA transgenic mice with a C57BL/6 N background carrying human wild type SNCA gene driven by the murine thymus cell antigen 1 promoter (6 months old) were purchased from Cyagen Biosciences. The present study did not involve sex-based study design, experiments and results.

## Mass spectra of the block strand in FNA-Fe$_3$O$_4$

FNA-Fe$_3$O$_4$ (100 μg mL$^{-1}$) was incubated with anticoagulated mice whole blood for 4 h at room temperature. Then FNA-Fe$_3$O$_4$ was magnetically separated and carried out strand displacement reaction in a solution containing 300 nM SNCA mRNA. The supernatant after the reaction was injected into ESI-MS (LTQ XL, Thermo Scientific) to analyze the mass spectra of DNA strand.

## Biodistribution and ex vivo imaging

To study the brain enrichment and biodistribution of the responsive artificial enzyme, C57BL/6 N mice (average weight of 25 g, 6 months old) were injected with 100 μL of fluorescence-on FNA-Fe$_3$O$_4$ (1 mg mL$^{-1}$ in 1× PBS) from tail vein. Then, mice were anaesthetized using an isoflurane vaporizer at various time periods (0, 30 min, 1, 2 and 4 h). The in vivo fluorescence imaging was performed on a small animal imaging system (PerkinElmer IVIS Lumina XRMS Series III, ex 620 nm & em 710 nm).

After the final in vivo fluorescence imaging, the main organs (brain, heart, liver, spleen, lung and kidney) of mice were carefully collected for ex vivo imaging. Uninjected mice were used as control. The mean fluorescence signals from different organs were analyzed using Living Image v4.7.2 software. Finally, brain tissue was preserved in 4% paraformaldehyde fixative for immunofluorescence analysis.

## Determination of in vivo blood circulation

Wild C57BL/6 N mice (6 months old) were received tail vein injection of FNA-Fe$_3$O$_4$ (1 mg mL$^{-1}$, 100 μL). At different time points (0.25, 0.5, 1, 2, 4, 8, 12, 16 and 24 h) post-injection, 50 μL of blood were collected through orbital sinus. The Fe concentration in serum was determined by ICP-MS.

## Immunofluorescence analysis of tyrosine hydroxylase in brain slices

Fixed brain tissue was paraffin-embedded and sectioned, and then dehydrated and washed. The section was blocked with BSA for 30 min, incubated with primary antibody in PBST overnight at 4 °C. The section was then washed 3 × 5 min in PBST, and incubated with secondary antibody in PBST for 1 h at room temperature. Then, the section was incubated sequentially with DAPI staining solution and auto-fluorescence quencher. Finally, after three washes with PBS, the section was mounted with anti-fluorescence quenching mounting medium, and the mounted section was observed and imaged under CLSM. Primary antibody against tyrosine hydroxylase (Servicebio, validated by manufacturer, cat# GB11181, 1:1000 dilution) and secondary antibody goat anti-rabbit labeled with FITC (Servicebio, validated by manufacturer, cat# GB22303, 1:100 dilution) were used.

## In vivo fluorescence imaging of SNCA mRNA by the responsive artificial enzyme

To investigate the in vivo fluorescence imaging function of the responsive artificial enzyme in PD mice, Thy1-SNCA transgenic and wild-type mice (6 months old) were injected with 100 μL of FNA-Fe$_3$O$_4$ (1 mg mL$^{-1}$ in 1× PBS). Then, mice were anesthetized using an isoflurane vaporizer at different time points (0, 1, 2 and 4 h). In vivo fluorescence imaging was performed on a small animal imaging system (PerkinElmer IVIS Lumina XRMS Series III, ex 620 nm & em 710 nm).

## In vivo microdialysis

Wild C57BL/6 N mice (6 months old) and Thy1-SNCA transgenic mice (6 months old) were anesthetized with sodium pentobarbital (50 mg kg$^{-1}$) and positioned in a stereotaxic frame (RWD Life Science Co., Ltd). The skull was exposed by a midline incision, and a 1.0-mm-diameter hole was drilled for placement of a microdialysis guide cannula (Eicom, Kyoto, Japan) into the striatum (anteroposterior: −0.46 mm; lateral: 1.3 mm; dorsoventral: 3.2 mm[50]). The guide cannula was fixed to the skull with cranioplastic cement.

The inflow line to the microdialysis probe was connected to a microinfusion pump via a quartzlined liquid swivel. A microdialysis

probe (2 mm membrane, 50 kDa cutoff) was inserted into the guide cannula to perfuse artificial cerebrospinal fluid (145 mM NaCl, 2.8 mM KCl, 1.2 mM CaCl$_2$, 1.2 mM MgCl$_2$, 0.25 mM AA, and 5.4 mM D-glucose) at a flow rate of 2 μL min$^{-1}$. After a 60 min equilibration period, perfusate samples were collected.

For artificial enzymes evoked Dopa and dopamine overflow, dialysate samples were collected up to 12 h. At 60 min, animals received a tail vein injection of artificial enzymes (4 mg kg$^{-1}$). For Dopa administration evoked dopamine overflow, dialysate samples were collected every 20 min up to 4 h. At 40 min, animals received intraperitoneal injections of Dopa (5, 10 or 15 mg kg$^{-1}$). Collected samples were stored at −80 °C prior to assay.

### Treatment process and behavioral tests
During the treatment, four groups of Thy1-SNCA transgenic mice (6 months old) received tail vein injection of PBS, ASO-Fe$_3$O$_4$, Cat-Fe$_3$O$_4$ and FNA-Fe$_3$O$_4$ every 4 days, respectively. A total of five injections were administered at the same dose (1 mg mL$^{-1}$, 100 μL) over a 20 days treatment period. Wild C57BL/6 N mice (6 months old) were used as healthy model controls. On the last day of the treatment period, behavioral tests were performed on each group of mice, including pole test, wire suspension test and passive avoidance test. Finally, mice were sacrificed. Cerebral hemispheres were used for immunoblot assay.

The pole test was used to assess dyskinesia in PD mouse models. A foam ball with a diameter of 2.5 cm was fixed on top of a wooden pole with length of 60 cm and diameter of 1 cm, and the lower end of the wooden pole was placed in a mouse cage. Mice were placed head-up on top of the foam ball, and the mice would turn down and crawl along the wooden pole to the ground. In experiment, the time for the mice to step on the wooden pole until it reached the ground was recorded as the descent time. Five trials were performed for each mouse with a 5 min interval.

The wire suspension test was used to assess muscle strength and grip reflex of mice. A standard wire cage lid was used. Duct tape was placed around the perimeter of the lid to prevent mice from walking off the edge. Mice were placed on top of the cage lid. The lid was slightly shaken to force the mice to grasp the wire, and was then turned upside down. The time for the mice remained suspended on the wire was recorded as the wire suspension time.

The step-through passive avoidance test was used to assess the learning and memory ability of mice. The passive avoidance apparatus (50 cm × 25 cm × 25 cm) consists of two compartments connected by a sliding door. The light intensity in the bright compartment was 300 lux, and the dark compartment was not illuminated. Mice roamed freely in the two compartments to acclimate to the environment, and were then trained and tested. During training, mice were placed in the bright compartment with sliding doors open. Once the mice entered the dark compartment, a weak electrical stimulus (0.2 mA) was delivered through the grid floor 2 s later, and the "step-through latency in training" was recorded. During the test phase 24 h later, the mice were placed in the bright compartment again and the latency to enter the dark compartment was measured as "step-through latency in 24 h retention", with a cut-off time of 300 s. No electrical stimulus was given during the test phase. Significantly prolonged step-through latency during the test phase indicates more intact learning and memory abilities.

### Serum biochemistry assays
Mice received tail vein injections of PBS, ASO-Fe$_3$O$_4$, Cat-Fe$_3$O$_4$ and FNA-Fe$_3$O$_4$ every 4 days at the same dose (1 mg mL$^{-1}$, 100 μL) over five injections. Three days after the final injection, the mice were sacrificed and their blood was collected. An automatic biochemical analyzer were used to obtain serum biochemical analysis data.

### Hemolysis assay
The red blood cells were collected from the mice whole blood. After washing with PBS, the red blood cells were treated with different concentrations of FNA-Fe$_3$O$_4$ (0–1000 μg mL$^{-1}$) at room temperature for 4 h. The negative and positive controls were treated with PBS and deionized water, respectively. After centrifugation, the absorbance of the supernatants was recorded at 540 nm.

### Statistics and reproducibility
The data are reported as mean ± SD. Statistical analysis was performed using Microsoft Excel 2016. Statistical analyses in all figures were performed by two-tailed Student's *t*-test. The significance level is *$P < 0.05$, **$P < 0.01$ and ***$P < 0.001$.

No statistical method was used to predetermine sample sizes. The sample sizes are consistent with those generally adopted and accepted in this field. No data were excluded from the analyses. All experimental samples and animal models were allocated randomly to each group. In animal experiments, the investigators were blinded in the process of group allocation and tail vain administration. In other experiments, the investigators were not blinded to allocation during experiments and outcome assessment.

### Reporting summary
Further information on research design is available in the Nature Portfolio Reporting Summary linked to this article.

## Data availability
All data generated that support the findings of this study are present in the main text and the Supplementary Information file. Source Data are provided with this paper.

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

## Acknowledgements

This work was supported by National Key R&D Program of China (2018YFA0902600, C.L.), National Natural Science Foundation of China (22174019, C.L.), the Natural Science Foundation of Fujian (2020J06036, C.L.), and the Major Project of Science and Technology of Fujian Province (2020HZ06006, H.Y.).

## Author contributions

X.F. and C.L. conceived the study, generated the hypotheses and designed the experiments. X.F. performed all the experiments and analysed the data. M.Y. participated in the cell culture and animal model operation. F.Z. and A.Y. participated in the synthesis and characterization of materials. Qia.L. performed the ESR experiments. S.L. and Xiny.W.

performed the in vivo microdialysis experiments. H.L. and Y.Y. performed the UPLC-MS measurements. Xin.W. participated in the western blot assays. Qit.L. participated in the PAGE analysis. X.F., C.L. and H.Y. wrote the manuscript. All authors read and approved the final version of the manuscript.

## Competing interests

The authors declare no competing interests.
