## [Peer Review File · Nature Communications]

Reviewers' Comments:

Reviewer #1:

Remarks to the Author:

This article reported a functional nucleic acid-based responsive artificial enzyme (FNA-Fe₃O₄) for in situ continuous Dopa production by crossing the blood-brain barrier. The nanomaterial could mimic tyrosine hydroxylase. Tyrosine aptamer allowed for specific binding of tyrosine substrates. Transferrin receptor aptamer was used to crossing the blood-brain barrier. SNCA mRNA antisense oligonucleotide was used for gene therapy and fluorescence imaging by strand displacement reactions. The nanomaterial integrated responsive tyrosine hydroxylase-mimicking activity, gene therapy, and fluorescence imaging of PD. The manuscript provided a paradigm for the advanced therapeutic strategy of PD.

Overall, the work is excellent. The overall structure of the article is complete, and the ideas are very novel. The data are well documented. This manuscript is strongly recommended for publication in NATURE COMMUNICATIONS. Some problems deserve to be addressed as follows:

1. DOPA is used for treatment for PD due to its ability to cross the blood-brain barrier. The nanomaterial could cross the blood-brain barrier for in situ continuous DOPA production. Why was the nanomaterial not used for in situ continuous and direct dopamine production rather than DOPA production. For example, tyramine could be used as the substrate for the mimic tyrosine hydroxylase to convert into dopamine. It was reported TyrH as well as nanozyme of TyrH both can catalyze the hydroxylation of tyramine. Figure 5I also supported the reaction for tyramine. The low conversion of tyramine in Figure 5I might be improved using tyramine aptamer as what did for Apt-Fe₃O₄.
2. The nanomaterial showed multifunction, including in situ continuous DOPA production, gene therapy, and fluorescence imaging. Please explain the significance and necessity for integration of such many functions in one material. In the word, how to avoid the mutual interference by each other from these three aspects?
3. The system was a little complex. FNA-Fe₃O₄ integrates responsive tyrosine hydroxylase-mimicking activity, gene therapy, and fluorescence imaging of PD. How about the contributions of Nanozyme and SNCA mRNA antisense on the in vivo biological effects of FNA-Fe₃O₄? Which one accounted for the predominant contribution? Further comparison and analysis between Cat-Fe₃O₄ and ASO-Fe₃O₄ would be encouraged.
4. Catalytic mechanism for the mimic tyrosine hydroxylase was studied. The manuscript declared the involvement of Fe(IV)=O on the basis of ·OOH. Can the manuscript provide some more direct and powerful experimental evidence for formation of Fe(IV)=O. If experimental data was lacked, it is suggested to change the solid arrow between ·OOH and Fe(IV)=O in Figure 2G to dashed arrow. And detail on generation of Fe(IV)=O from ·OOH should be described based on references.
5. Radicals were investigated by ESR. Whether other method also applied, such as quenching experiments using scavengers. Especially for the deduction "The ·OH produced by the reaction of Fe₃O₄ with H₂O₂ was converted into ·OOH under the catalysis of AA."
6. Figure 2G. The dashed circle referred to the intermediate and process of hydroxylation of tyrosine to DOPA. Fe^{III} was included in that scheme. However, it seems that the mechanism for TyrH does not include Fe^{III}-O⁻.
7. What was the tyrosine concentration in Figure 3H? It should be provided in the caption. The information and value for yield or conversion was lacked in the text. It should be mentioned in the text. From Figure 2I and Figure S6, the yield or conversion seemed less than 5%. Comparison with previous works should be conducted. In some recent publications (ACS APPL NANO MATER, 2022; ChemNanoMat, 2022), Shang and coworkers described the utilization of Fe₃O₄ for production of DOPA as nanozyme for Tyrosine Hydroxylase mimics. Please compare with them in both the Introduction Section and Results Section.
8. The authors might calculate Km for Figure 3B. It seemed that Vmax for Apt-Fe₃O₄ was larger

than Fe₃O₄, but K_m for Apt-Fe₃O₄ was close to Fe₃O₄. If aptamer for Apt-Fe₃O₄ was intended to be used as substrate-specific binding components, a decreased K_m was expected. Please explain the reason.

9. What was the role for BHQ2? Why was BHQ2 not used in Figure S15?

10. How to avoid that antisense RNA was displaced by other mRNA instead of SNCA mRNA in vivo?

11. Both 5 mM ascorbic acid and 5 mM H₂O₂ was used for mimic tyrosine hydroxylase to convert into DOPA. Acceptable yield was obtained. How to realize such high concentration in vivo application? How much is the ascorbic acid and H₂O₂ concentration in brain? The catalytic performance under ascorbic acid H₂O₂ at concentration identical with brain might be test.

12. Ascorbic acid and H₂O₂ was used. Did ascorbic acid and H₂O₂ interfere fluorescence imaging for FNA-Fe₃O₄? For example, how about fluorescence quenching under ascorbic acid and H₂O₂ was used?

13. The manuscript investigated the SNCA protein level upon FNA-Fe₃O₄. How about the mRNA level?

14. Figure 6B and 6C, why was control for PD mice without any treatment not included? PD mice treated by PBS or other nanomaterial might give more convincing evidence.

Reviewer #2:

Remarks to the Author:

In this manuscript, the authors designed and engineered a functional nucleic acid-based responsive artificial enzyme (FNA-Fe₃O₄) for in situ continuous Dopa production. FNA-Fe₃O₄ could not only cross the blood-brain barrier and target diseased neurons depending on transferrin receptor aptamer, but also responded to overexpressed α -synuclein mRNA in diseased neurons for antisense oligonucleotide therapy and fluorescence imaging, while converting to tyrosine aptamer-based artificial enzyme that mimicked tyrosine hydroxylase for in situ continuous Dopa production. In my opinion, this research is well-organized and thoroughly carried out. I would like to recommend acceptance of this work after the following issues are addressed:

1. How did the functional nucleic acid-based responsive artificial enzyme achieve fluorescence imaging? What is the significance of fluorescence imaging during the treatment of Parkinson's disease?

2. Why did the authors select the tyrosine aptamer as the substrate binding site? How about the advantages and/or disadvantages of aptamers as substrate-specific binding components compared to antibodies?

3. In Figure 5d, why did the Dopa incubation induce dramatic fluctuations in intracellular dopamine levels?

4. In Figure 6a and Supplementary Fig. 19, why did mice have apparent fluorescent signals in their limbs? Besides, how about the in vivo brain accumulation in mice injected with fluorescence-on FNA-Fe₃O₄ after more than 4h?

5. Since the biosafety of biomaterials is a prerequisite for their further clinical application, how about the in vivo biosafety of FNA-Fe₃O₄ after the intravenous injection?

6. How about the half-life of blood circulation and retention time of FNA-Fe₃O₄ in vivo? Why did the authors adopt tail vein injection of FNA-Fe₃O₄ every 4 days?

7. In the Introduction section, more pioneering work and progress need to be cited when the author describes the field of treatment of brain disease, such as Exploration 2022, 20210274;

Reviewer #3:

Remarks to the Author:

In this study the investigators propose a very sophisticated and clever approach that could represent a new avenue for the symptomatic, and perhaps, disease modifying treatment of Parkinson's disease. The symptomatic approach is based on an artificial enzyme - a nanozyme - that could produce steady levels of dopamine in the brain in situ. The disease modifying approach involves gene therapy using an antisense mRNA oligonucleotide aimed at decreasing intracellular alpha synuclein (SNCA) production and accumulation, which has a presumably toxic effect for neurons.

The authors investigated a Fe₃O₄ nanozyme that could mimic the function of the tyrosine hydroxylase enzyme for dopa synthesis (a dopamine precursor) in the presence of hydrogen peroxide H₂O₂ and ascorbic acid. For in vivo applications, the authors modified the core structure of the Fe₃O₄ nanozyme as follows:

A) The authors constructed a tyrosine aptamer-based nanozyme (Apt-Fe₃O₄) to promote substrate capture despite the crowded intracellular macromolecular environment.

B) To specifically target pathological neurons in the brain and to avoid uptake in other neurons (potentially leading to side effects) the investigators performed hybridization between the Apt-Fe₃O₄ and a block strand containing α -synuclein (SNCA) mRNA antisense oligonucleotide and transferrin receptor (TfR) aptamer under two assumptions:

- the resulting molecular construct would cross the blood brain barrier in vivo and enter the intracellular neuronal compartment through receptor-mediated transport implicating the transferrin receptor

- the catalytic activity of the nanozyme in the cell would be enabled by strand displacement triggered by high intracellular SNCA mRNA levels that are specifically found in pathological neurons.

C) Furthermore, the aforementioned strand displacement reactions is expected to promote gene therapy by the release of the SNCA mRNA antisense oligonucleotide, opening the possibility of a disease-modifying effect

These hypotheses seem to be tested in well-designed complex experiments that included appropriate controls. The experimental procedures could be divided into 3 parts:

- construction and characterization of FNA-Fe₃O₄

- in vivo cellular experiments showed that the intracellular dopamine content gradually raised & reached a plateau within 8 h without evidence for cellular toxicity

- in vivo animal experiments after injection in the tail vein in wild C57BL/6N mice, and in Thy1-SNCA transgenic mice (which carries human wild type SNCA driven by the murine thymus cell antigen 1 (Thy1) promoter) used as normal and PD animal models, respectively, provided evidence for continuous in situ dopa production as shown by striatal microdialysis and mitigation of aberrant SNCA overexpression in transgenic mice. At the behavioural level, this was associated with improved motor performance in the pole climbing test and the wire suspension test. Biodistribution analyses showed that FNA-Fe₃O₄ accumulated in large amounts in the liver, and in small amounts in the kidney, lung and spleen. Cell viability test results did not show evidence for FNA-Fe₃O₄ cytotoxicity.

Despite the complexity of the proposed methodological approach, the manuscript is well written with appropriate figures to support the results. The methods are sound although I'm not an expert in molecular biology. Overall, if the methods could be reproduced in the future, the approach hold great promise for the treatment of Parkinson's disease. One important aspect to address in the near future concerns detailed toxicity assessment of the nanozyme in normal and pathological individuals.

My recommendation is : accept the manuscript with minor comments on potentiel whole body toxicity to be adressed in the discussion section

Reviewer #4:

Remarks to the Author:

In this manuscript, the authors described a method to treat Parkinson's disease. They designed the functional nucleic acid-based responsive artificial enzyme FNA-Fe₃O₄ exhibited efficient tyrosine hydroxylase-mimicking activity. While this work appears to be of some interest, I suggest rejecting this manuscript at least in its current form. The main reasons are: 1) the construction of DNA-modified Fe₃O₄ is not novel; 2) it's surprising that DNA aptamers survive in blood circulation and then penetration BBB to reach the brain. I also request the authors to consider the following concerns.

1. The authors claimed that "Fe₃O₄ nanozymes were covalently conjugated with the 5'-terminal amino-modified tyrosine aptamer through EDC/NHS mediated amide condensation reaction to construct the tyrosine aptamer-based artificial enzyme Apt-Fe₃O₄" in page 5. Does DNA modification with different densities affect the catalytic efficiency of artificial enzyme? The authors needed to calculate how many strands of DNA were modified on each nanoparticle.
2. The authors described that "we proved that Fe₃O₄ could generate Fe(IV)=O active intermediates" in page 11. The authors should provide more data to support this claim.
3. The level of dopamine secreted by PD cells treated with FNA-Fe₃O₄ is much higher than normal cells in Fig. 5c. Whether the treatment will affect the homeostasis of cells?
4. In Fig. 6g, why was there almost no therapeutic effect in group ASO-Fe₃O₄?
5. In addition to BV2 cells, the authors should also test the cytotoxicity of the artificial enzyme to other related cells.
6. The format of references needs to be uniform.

Response to Reviewers' Comments

Reviewer #1 (Remarks to the Author):

This article reported a functional nucleic acid-based responsive artificial enzyme (FNA-Fe₃O₄) for in situ continuous Dopa production by crossing the blood-brain barrier. The nanomaterial could mimic tyrosine hydroxylase. Tyrosine aptamer allowed for specific binding of tyrosine substrates. Transferrin receptor aptamer was used to crossing the blood-brain barrier. SNCA mRNA antisense oligonucleotide was used for gene therapy and fluorescence imaging by strand displacement reactions. The nanomaterial integrated responsive tyrosine hydroxylase-mimicking activity, gene therapy, and fluorescence imaging of PD. The manuscript provided a paradigm for the advanced therapeutic strategy of PD.

Overall, the work is excellent. The overall structure of the article is complete, and the ideas are very novel. The data are well documented. This manuscript is strongly recommended for publication in NATURE COMMUNICATIONS.

Response:

Thanks for the reviewer's positive comments and suggestions. We have revised the manuscript accordingly and given detailed replies point by point.

Some problems deserve to be addressed as follows:

1. DOPA is used for treatment for PD due to its ability to cross the blood-brain barrier. The nanomaterial could cross the blood-brain barrier for in situ continuous DOPA production. Why was the nanomaterial not used for in situ continuous and direct dopamine production rather than DOPA production. For example, tyramine could be used as the substrate for the mimic tyrosine hydroxylase to convert into dopamine. It was reported TyrH as well as nanozyme of TyrH both can catalyze the hydroxylation of tyramine. Figure 5I also supported the reaction for tyramine. The low conversion of tyramine in Figure 5I might be improved using tyramine aptamer as what did for Apt-Fe₃O₄.

Response:

Thanks to the reviewer's comments. We agree with the reviewer that tyramine can also be catalyzed to produce dopamine. However, tyramine is a kind of "trace amines" in the body and the physiological concentration of tyramine (<10 ng g⁻¹; 100 nM) makes it unsuitable as conventional substrate source of dopamine. The level of tyramine is at least two orders of magnitude lower than that of the corresponding neurotransmitter (*Pharmacological Reviews*, 2018 (70), 549-620. DOI: 10.1124/pr.117.015305). Therefore, it is difficult to produce enough dopamine to activate nerve signals by using tyramine as hydroxylated substrate. In contrast, tyrosine is the major starting material for dopamine biosynthesis in vivo, thus this work chose to construct the artificial enzyme with tyrosine as a specific substrate.

2. The nanomaterial showed multifunction, including in situ continuous DOPA production, gene therapy, and fluorescence imaging. Please explain the significance and necessity for integration of such many functions in one material. In the word, how to avoid the mutual interference by each other from these three aspects?

Response:

Thanks to the reviewer's useful comments.

The two pathological features of PD are the inactivation of tyrosine hydroxylase in neurons and the overexpression of SNCA: (i) The inactivation of tyrosine hydroxylase leads to the reduction of Dopa and dopamine synthesis, resulting in motor and mental symptoms of patients. Dopa supply alone can alleviate the clinical manifestations of PD, but it cannot reverse the PD pathological process; (ii) The overexpression of SNCA induces further degeneration and death of neurons, which aggravates the condition of PD. Gene therapy toward SNCA alone can retard the further deterioration of PD, but it cannot quickly improve the dyskinesia of patients. Therefore, the combination of in situ Dopa production and gene therapy toward SNCA is a more effective strategy for PD treatment. In addition, overexpressed SNCA mRNA is often used as a biomarker for evaluating the PD pathological process. Fluorescence imaging of SNCA mRNA can provide reference in the diagnosis and treatment of PD. Thus, the artificial enzyme FNA-Fe₃O₄, which integrates responsive tyrosine hydroxylase-mimicking activity, gene therapy, and fluorescence imaging of PD, may provide a broader prospect for the diagnosis and treatment of PD.

In the artificial enzyme design of this study, FNA-Fe₃O₄ undergoes strand displacement reaction response to SNCA mRNA. Fluorescence imaging is triggered by the separation of Cy5 and BHQ2 in FNA-Fe₃O₄ functional nucleic acid structure. Gene therapy is triggered by the complementary combination of block strand and SNCA mRNA. The artificial enzyme is transformed to Apt-Fe₃O₄ for in situ catalysis of Dopa production. The three aspects are independent, so they will not interfere with each other.

3. The system was a little complex. FNA-Fe₃O₄ integrates responsive tyrosine hydroxylase-mimicking activity, gene therapy, and fluorescence imaging of PD. How about the contributions of Nanozyme and SNCA mRNA antisense on the in vivo biological effects of FNA-Fe₃O₄? Which one accounted for the predominant contribution? Further comparison and analysis between Cat-Fe₃O₄ and ASO-Fe₃O₄ would be encouraged.

Response:

Thanks for the reviewer's thoughtful suggestions. Cat-Fe₃O₄ and ASO-Fe₃O₄ partially restored motor performance of PD mice in the pole climbing test and wire suspension test. The therapeutic effect of Cat-Fe₃O₄ alone was slightly better than that of ASO-Fe₃O₄, because in situ Dopa supply can directly act on dopaminergic nervous system. The contribution of antisense oligonucleotide therapy was to alleviate the PD pathological process. The FNA-Fe₃O₄ treated mice exhibited better performance than monotherapy, in which in situ Dopa supply was the main contribution to the therapeutic effect (Fig. 6g,h in the revised manuscript). Cat-Fe₃O₄ and ASO-Fe₃O₄ helped to improve the learning and memory of PD mice in passive avoidance test. In situ Dopa supply and antisense oligonucleotide therapy contributed to the therapeutic effect at a similar level (Fig. 6i in the revised manuscript).

In the revised manuscript, we discussed the contribution of Cat-Fe₃O₄ and ASO-Fe₃O₄ to the treatment of PD in behavioral tests (Page 12, left column, lines 35-41, 46-48).

4. Catalytic mechanism for the mimic tyrosine hydroxylase was studied. The manuscript declared the involvement of Fe(IV)=O on the basis of ·OOH. Can the manuscript provide some more direct and powerful experimental evidence for formation of Fe(IV)=O. If experimental data was lacked, it is suggested to change the solid arrow between ·OOH and Fe(IV)=O in Figure 2G to dashed arrow. And detail on generation of Fe(IV)=O from ·OOH should be described based on references.

Response:

Thanks for the reviewer's constructive suggestions. Following the suggestion, Fe(IV)=O in the reaction system was qualitatively detected and the results were added to the revised manuscript. Dimethyl sulfoxide (DMSO) is used as a probe for Fe(IV)=O, which reacts with Fe(IV)=O through oxygen atom transfer to produce dimethyl sulfone (MSM) (*Angew. Chem. Int. Ed.* 2005 (44), 6871-6874. DOI: 10.1002/anie.200502686). According to the GC-MS analysis results in Fig. 2i in the revised manuscript, the peak of MSM gradually increased with reaction time, suggesting the occurrence of Fe(IV)=O. The corresponding discussion was added (Page 5, left column, lines 38-45). In addition, we added the details on generation of Fe(IV)=O from ·OOH in Fig. 2j of the revised manuscript according to the references.

5. Radicals were investigated by ESR. Whether other method also applied, such as quenching experiments using scavengers. Especially for the deduction "The ·OH produced by the reaction of Fe₃O₄ with H₂O₂ was converted into ·OOH under the catalysis of AA."

Response:

Thanks for the reviewer's useful suggestion. In the revised manuscript, we further verified the active species in the reaction system through the quenching experiments using scavengers of ·OH and Fe(IV)=O. Tert-butanol tends to mainly quench ·OH, and ethanol is known to effectively scavenge both ·OH and Fe(IV)=O (*Chem. Eng. J.*, 2020 (382), 123013. DOI: 10.1016/j.cej.2019.123013). The addition of tert-butanol in the reaction system mildly inhibited the reaction by 39.4% after 5 min, while the addition of ethanol induced more serious inhibition by 60.4% (Fig. 2c in the revised manuscript). These differences could be attributed to the participation of ·OH and high-spin Fe(IV)=O in hydroxylation. The corresponding discussion was added (Page 5 left, lines 3-12).

6. Figure 2G. The dashed circle referred to the intermediate and process of hydroxylation of tyrosine to DOPA. Fe(III) was included in that scheme. However, it seems that the mechanism for TyrH does not include Fe(III)-O-

Response:

Thanks to the reviewer's comment. Through an extensive literature review on the reaction between Fe(IV)=O and tyrosine, we find that there is an indispensable intermediate Fe(III)-O-Ar, which is generated through the 1 e⁻ oxidation process of Fe(IV)=O (*Angew. Chem. Int. Ed.* 2021 (60), 20991-20998. DOI: 10.1002/anie.202108309; *J. Am. Chem. Soc.* 2007 (129), 11334-11335. DOI: 10.1021/ja074446s). Furthermore, it is also reported that Fe(III)-O-Ar will further produce thermodynamically more stable Fe(II)-O-Ar(+) through electron transfer process. Therefore, in the revised manuscript, we changed the Fe(III)-O-Ar in Fig. 2j to the more widely recognized Fe(II)-O-Ar(+).

7. What was the tyrosine concentration in Figure 3H? It should be provided in the caption. The information and value for yield or conversion was lacked in the text. It should be mentioned in the text. From Figure 2I and Figure S6, the yield or conversion seemed less than 5%. Comparison with previous works should be conducted. In some recent publications (ACS APPL NANO MATER, 2022; ChemNanoMat, 2022), Shang and coworkers described the utilization of Fe₃O₄ for production of DOPA as nanozyme for Tyrosine Hydroxylase mimics. Please compare with them in both the Introduction Section and Results Section.

Response:

Thanks for the reviewer's thoughtful suggestions. In the revised manuscript, we provided a more detailed description in the caption of Fig. 3h and 3i.

In industrialized production, the yield or conversion rate is the meaningful index to measure the production efficiency of reaction system. However, for the in vivo application of artificial enzymes, the final yield or conversion rate is of little significance. Because in the physiological environment, reactants and cofactors (tyrosine, ascorbic acid, H₂O₂) are under the regulation of organisms, and their concentrations are basically stable. Therefore, the catalytic performance is suitable to be compared by the initial reaction rate or the conversion rate in a very short time. At this time, the concentrations of reactants and cofactors have little change. Fig. 3i and Supplementary Fig. 5 (that is, Supplementary Fig. 6 in the original manuscript) show the conversion rate of the reaction at 5 min, so the conversion rate is low. In the revised manuscript, we added a description of the reaction time (Page 7, right column, lines 20-21, 30-31).

We thank the reviewer for reminding us of the relevant references on the reaction rate and conversion rate of tyrosine hydroxylation catalyzed by Fe₃O₄. We added these relevant references in the Introduction section and Discussion section (Page 2, left column, lines 35-38, ref. 24-25; Page 12, right column, lines 36-39, ref. 24-25).

In addition, the properties of artificial enzyme can be evaluated more appropriately by using the parameters that reflect the properties of enzymatic reaction: Michaelis-Menten constant (K_m) and the reaction rate when the enzyme is saturated with substrate (V_{max}). In the revised manuscript, we added the corresponding parameters (Supplementary Table 2) in Fig. 3b,c,f,g, so that we can understand the nature of enzymatic reaction more conveniently.

8. The authors might calculate K_m for Figure 3B. It seemed that V_{max} for Apt-Fe₃O₄ was larger than Fe₃O₄, but K_m for Apt-Fe₃O₄ was close to Fe₃O₄. If aptamer for Apt-Fe₃O₄ was intended to be used as substrate-specific binding components, a decreased K_m was expected. Please explain the reason.

Response:

Thanks to the reviewer's comment and reminder. In Fig. 3b of the original manuscript, the horizontal coordinate range of Michaelis-Menten-type kinetic curves does not reach the saturated substrate concentration of Fe₃O₄. The initial reaction rate of Fe₃O₄ saturated with substrate is much higher than that in 2000 uM tyrosine. In this case, it is hard to visually observe the K_m value of Fe₃O₄ according to the curve in Fig. 3b. Therefore, in the revised manuscript, we added Supplementary Table 2, which is used to digitally present the parameters reflecting the properties of the enzymatic reaction in Fig. 3b,c,f,g: Michaelis-Menten constant (K_m) and the reaction rate when the enzyme is saturated with substrate (V_{max}). As shown in Supplementary Table 2, the K_m value of Apt-Fe₃O₄ is significantly smaller than that of Fe₃O₄, indicating that Apt-Fe₃O₄ has higher affinity for substrate.

9. What was the role for BHQ2? Why was BHQ2 not used in Figure S15?

Response:

Thanks to the reviewer's comments. The fluorescence quenching pairs of Cy5 and BHQ2 were used for fluorescence imaging of SNCA mRNA. In the double-strand DNA structure of FNA-Fe₃O₄, BHQ2 would quench the nearby fluorescence of Cy5. After strand displacement reaction with SNCA mRNA, BHQ2 was separated from Cy5, thus generating turn-on fluorescence signal for fluorescence imaging (Supplementary Fig. 7a). The relevant discussion was added (Page 7, left column, lines 29-32).

The fluorescence-on FNA-Fe₃O₄ in Supplementary Fig. 17 was used to study the distribution of FNA-Fe₃O₄ in cells or in vivo, and it was necessary to keep the fluorescence signal of FNA-Fe₃O₄ always observable. Therefore, in Fig. 4c,d and Supplementary Fig. 27, 29 and 30, we used the fluorescence-on FNA-Fe₃O₄ without BHQ2. The relevant discussion has been mentioned in the manuscript (Page 8, left column, lines 19-21).

10. How to avoid that antisense RNA was displaced by other mRNA instead of SNCA mRNA in vivo?

Response:

Thanks to the reviewer's comment. Antisense oligonucleotide (ASO) is a segment of nucleic acid complementary to a certain region of target gene or mRNA, which can bind to the target gene/mRNA to inhibit the expression of the gene/mRNA. In our work, the designed ASO fragment only bound to SNCA mRNA under the guidance of the principle of base complementary pairing, and it was difficult to bind to other mRNAs. In the manuscript, the reaction specificity with other analogous mRNAs was studied (Supplementary Fig. 12), which proved that FNA-Fe₃O₄ had good reaction specificity towards SNCA mRNA.

11. Both 5 mM ascorbic acid and 5 mM H₂O₂ was used for mimic tyrosine hydroxylase to convert into DOPA. Acceptable yield was obtained. How to realize such high concentration in vivo application? How much is the ascorbic acid and H₂O₂ concentration in brain? The catalytic performance under ascorbic acid H₂O₂ at concentration identical with brain might be test.

Response:

Thanks for the reviewer's insightful comments. Before the catalytic performance test of this work, we investigated the concentrations of ascorbic acid (AA) and H₂O₂ in the brain in advance. In most neurons, AA is maintained at a high concentration, usually 2-10 mM (*Free Radical Bio. Med.* 2009 (46), 719-730. DOI: 10.1016/j.freeradbiomed.2008.12.018; *BBA-Rev. Cancer* 2012 (1826), 443-457. DOI: 10.1016/j.bbcan.2012.06.003). H₂O₂ is at a low level in normal neurons, but it is overproduced in Parkinson's diseased-neurons (*Prog. Neurobiol.* 2016 (147), 1-19. DOI: 10.1016/j.pneurobio.2016.07.005; *Neurology* 1996 (47), S161-S170. DOI: 10.1212/WNL.47.6_Suppl_3.161S). Although it is difficult to know the specific concentration of H₂O₂ in Parkinson's diseased-neurons, 5 mM H₂O₂ is widely used as a method to simulate high-level oxidative stress in vitro (*Small Methods* 2019 (3), 1900013. DOI: 10.1002/smt.201900013; *J Mater. Chem. B* 2015 (3), 1597-1604. DOI: 10.1039/c4tb01709a).

Therefore, we chose 5 mM AA and 5 mM H₂O₂ to simulate the physiological concentrations in the brain of Parkinson's disease. When the artificial enzyme was used in vivo, endogenous physiological concentrations of AA and H₂O₂ were also in the order of mM, which was enough to support the catalytic reaction. In the revised manuscript, we added two references (Page 3, left column, lines 10-11, ref. 37-38) to illustrate the concentrations of AA and H₂O₂ in brain.

12. Ascorbic acid and H₂O₂ was used. Did ascorbic acid and H₂O₂ interfere fluorescence imaging for FNA-Fe₃O₄? For example, how about fluorescence quenching under ascorbic acid and H₂O₂ was used?

Response:

Thanks for the reviewer's constructive suggestion. Following the suggestion, we measured the fluorescence signal responsivity of FNA-Fe₃O₄ in the presence of AA and H₂O₂ (Supplementary Fig. 11 in the revised manuscript). AA and H₂O₂ partially quenched the fluorescence signal of Cy5, but FNA-Fe₃O₄ still maintained a good response to SNCA mRNA to turn on the fluorescence signal. The relevant discussion was added in the revised manuscript (Page 7, left column, lines 42-45).

13. The manuscript investigated the SNCA protein level upon FNA-Fe₃O₄. How about the mRNA level?

Response:

Thanks for the reviewer's thoughtful reminder. The SNCA mRNA levels of SH-SY5Y cells after artificial enzymes co-incubation were explored and the results were added in the revised manuscript (Supplementary Fig. 22). After 36 h of artificial enzymes co-incubation, FNA-Fe₃O₄ and ASO-Fe₃O₄ reduced the levels of SNCA mRNA in PD cell model to about 70%. The relevant discussion was added (Page 10, left column, lines 36-40).

14. Figure 6B and 6C, why was control for PD mice without any treatment not included? PD mice treated by PBS or other nanomaterial might give more convincing evidence.

Response:

Thanks for the reviewer's thoughtful suggestion. Following the suggestion, the microdialysis results of PD mice injected with PBS were measured and added to Fig. 6b and 6c in the revised manuscript. The contents of Dopa and dopamine in striatum of PBS injected PD mice tended to be stable within 12 h, which was obviously different from those in FNA-Fe₃O₄ and Cat-Fe₃O₄ treatment groups.

Reviewer #2 (Remarks to the Author):

In this manuscript, the authors designed and engineered a functional nucleic acid-based responsive artificial enzyme (FNA-Fe₃O₄) for in situ continuous Dopa production. FNA-Fe₃O₄ could not only cross the blood-brain barrier and target diseased neurons depending on transferrin receptor aptamer, but also responded to overexpressed α -synuclein mRNA in diseased neurons for antisense oligonucleotide therapy and fluorescence imaging, while converting to tyrosine aptamer-based artificial enzyme that mimicked tyrosine hydroxylase for in situ continuous Dopa production. In my opinion, this research is well-organized and thoroughly carried out.

Response:

Thanks for the reviewer's positive comments and suggestions. We have revised the manuscript accordingly and given detailed replies point by point.

I would like to recommend acceptance of this work after the following issues are addressed:
1. How did the functional nucleic acid-based responsive artificial enzyme achieve fluorescence imaging? What is the significance of fluorescence imaging during the treatment of Parkinson's disease?

Response:

Thanks to the reviewer's comments and reminders.

(1) Achievement of fluorescence imaging: Cy5 and BHQ2 were modified on the functional nucleic acid double-strand structure of FNA-Fe₃O₄ as fluorescence quenching pairs, as shown in Supplementary Fig. 7a. BHQ2 would quench the nearby fluorescence of Cy5. After reached neurons and was internalized, FNA-Fe₃O₄ would respond to overexpressed SNCA mRNA in Parkinson's diseased-neurons to undergo strand displacement reaction. After strand displacement reaction, BHQ2 was separated from Cy5, thus generating turn-on fluorescence signal for fluorescence imaging. By evaluating the signal intensity of fluorescence imaging, the expression level of SNCA mRNA in neurons can be preliminarily judged. The relevant discussion was added (Page 7, left column, lines 28-32).

(2) Significance of fluorescence imaging of PD: The degenerative death of neurons in the pathological process of Parkinson's disease (PD) is difficult to reverse. Early detection of preclinical patients can prevent the occurrence and progress of the disease earlier, which is of great significance to the treatment of PD. SNCA mRNA is one of the biomarkers of PD, which is overexpressed in neurons of patients with PD. Fluorescence imaging of the expression level of SNCA mRNA can preliminarily evaluate the progress of PD and provide guidance for the treatment of PD. The relevant discussion has been mentioned in the manuscript (Page 2, right column, lines 31-35).

2. Why did the authors select the tyrosine aptamer as the substrate binding site? How about the advantages and/or disadvantages of aptamers as substrate-specific binding components compared to antibodies?

Response:

Thanks for the reviewer's insightful comments.

(1) The reason for selecting tyrosine aptamer: One of the pathological features of PD is dopamine deficiency. The hydroxylation of tyrosine to produce Dopa is the rate-limiting step for neurons to synthesize dopamine. The present artificial enzyme in our work can simulate

tyrosine hydroxylase, catalyzing the hydroxylation of tyrosine substrate. Therefore, we select the tyrosine aptamer as the substrate binding site. Tyrosine aptamer can specifically bind tyrosine, thus enriching tyrosine and increasing the local concentration of tyrosine. The increase in local concentration of tyrosine greatly increases the catalytic activity and selectivity of the artificial enzyme to tyrosine, which makes it feasible to apply the artificial enzyme in vivo.

(2) Advantages and disadvantages of aptamers: Antibodies generally have stronger binding abilities to substrates than aptamers. However, antibodies have large sizes, which makes it difficult for the substrate to approach the catalytic active interface. And antibody-modified nanoparticles have difficulties in penetrating the blood-brain barrier. More importantly, it is difficult for antibodies to perform responsive conformational switching to control the catalytic activity of artificial enzymes. Compared with antibodies, aptamers can be additionally functionalized according to their nucleic acid sequences and realize controllable structural changes under the guidance of the principle of base complementary pairing. In this work, we designed the block strand that hybridized with tyrosine aptamer. Transferrin receptor aptamer extended from the block strand enabled FNA-Fe₃O₄ to cross the blood-brain barrier. Moreover, the conformations of tyrosine aptamers can be switched through hybridization and dehybridization, thus achieving control of the catalytic activity of artificial enzymes. Therefore, in this work, aptamer is a better choice than antibody in the construction of the artificial enzyme.

3. In Figure 5d, why did the Dopa incubation induce dramatic fluctuations in intracellular dopamine levels?

Response:

Thanks to the reviewer's comment. In the cell experiment of Fig. 5d, we simulated the rapid in vivo clearance of Dopa. Therefore, the cells were first incubated with the culture medium containing Dopa for 2 h. During this period, Dopa was uptaken by cells and converted into dopamine under the catalysis of Dopa decarboxylase, thus showing a rapid increase in intracellular dopamine levels. After 2 h, the medium was replaced to fresh culture medium without Dopa, which was used to simulate the short half-life of Dopa administration in vivo. The cells no longer got Dopa supply, so the dopamine level gradually decreased after 2 h. Overall, the short-term Dopa incubation induced dramatic fluctuations of intracellular dopamine levels within 24 h. The experimental procedures were described in more detail (page 15, left column, lines 2-5), and the relevant discussion has been added in the manuscript (Page 10, left column, lines 12-17).

4. In Figure 6a and Supplementary Fig. 19, why did mice have apparent fluorescent signals in their limbs? Besides, how about the in vivo brain accumulation in mice injected with fluorescence-on FNA-Fe₃O₄ after more than 4h?

Response:

Thanks for the reviewer's constructive suggestion.

Autofluorescence exists in organisms. Raising the lower intensity limit of in vivo fluorescence imaging can remove weak autofluorescence signals, but some details of fluorescence imaging will be lost at the same time. The principle of our processing of living fluorescence images is to keep reasonable fluorescence signal points as much as possible, thus the fluorescence signals of mouse limbs are observable.

In the revised manuscript, we re-collected the living fluorescence images of mice within 24 h after injection of fluorescence-on FNA-Fe₃O₄, replacing the original 4 h images (Supplementary Fig. 27, that is Supplementary Fig. 19 in the original manuscript). FNA-Fe₃O₄ gradually accumulated in the brain after injection, reached the peak after 4 h and remained in the brain within 24 h.

5. Since the biosafety of biomaterials is a prerequisite for their further clinical application, how about the in vivo biosafety of FNA-Fe₃O₄ after the intravenous injection?

Response:

Thank the reviewer's thoughtful comment. During the literature investigation before the experiment, we learned that Fe₃O₄ nanoparticles were approved by the U.S. Food and Drug Administration for use in biomedical fields and have been widely used in magnetic resonance imaging, targeted drug carriers and tissue engineering. Nucleic acids, as one of the components of living body, have been widely recognized for their biocompatibility. Therefore, FNA-Fe₃O₄ constructed by Fe₃O₄ nanoparticles and functional nucleic acids should have good biosafety in theory.

In the original manuscript, we characterized the Dopa contents and dopamine contents in peripheral tissues (Supplementary Fig. 35 and 36), pathological sections of main organs (Supplementary Fig. 37), and body weight (Supplementary Fig. 40) of mice after intravenous administration of FNA-Fe₃O₄. The corresponding results confirmed that FNA-Fe₃O₄ had good biosafety.

In the revised manuscript, we carried out the hemolytic test of FNA-Fe₃O₄ on red blood cells (Supplementary Fig. 39) and the serum biochemical assay after intravenous administration of FNA-Fe₃O₄ (Supplementary Fig. 38). The corresponding results were added in the revised manuscript and supplemented the good biosafety of FNA-Fe₃O₄. The relevant discussion was added (Page 12, right column, lines 13-21).

6. How about the half-life of blood circulation and retention time of FNA-Fe₃O₄ in vivo? Why did the authors adopt tail vein injection of FNA-Fe₃O₄ every 4 days?

Response:

Thanks to the reviewer's comments and reminders. Following the suggestion, the in vivo blood circulation curve of FNA-Fe₃O₄ was explored and added in the revised manuscript (Supplementary Fig. 28). The content of FNA-Fe₃O₄ in blood decreased gradually, and the half-life in blood circulation was about 1.2 h. In addition, according to Supplementary Fig. 27 in the revised manuscript, FNA-Fe₃O₄ gradually accumulated in the brain after injection, and reached the peak after 4 h. Even after 24 h, FNA-Fe₃O₄ still retained in the brain to a considerable extent. The relevant discussion was added (Page 10, right column, lines 21-27).

The frequency of tail vein injection once every 4 days was determined according to the retention of FNA-Fe₃O₄ in the brain, the therapeutic effect of FNA-Fe₃O₄ and the conventional operation methods in this field (*J. Am. Chem. Soc.* 2020 (142), 3862-3872. DOI: 10.1021/jacs.9b11490). This injection frequency could provide a good therapeutic effect and maintain the normal living habits of mice.

7. In the Introduction section, more pioneering work and progress need to be cited when the author describes the field of treatment of brain disease, such as Exploration 2022, 20210274; Nature Communications 2021,12, 2203.

Response:

Thanks for the reviewer's suggestion. The mentioned references described the work related to the treatment of brain diseases, thus we added these relevant references in the introduction section (ref. 17 and ref. 31).

Reviewer #3 (Remarks to the Author):

In this study the investigators propose a very sophisticated and clever approach that could represent a new avenue for the symptomatic, and perhaps, disease modifying treatment of Parkinson's disease. The symptomatic approach is based on an artificial enzyme - a nanozyme - that could produce steady levels of dopamine in the brain in situ. The disease modifying approach involves gene therapy using an antisense mRNA oligonucleotide aimed at decreasing intracellular alpha synuclein (SNCA) production and accumulation, which has a presumably toxic effect for neurons.

The authors investigated a Fe₃O₄ nanozyme that could mimic the function of the tyrosine hydroxylase enzyme for dopa synthesis (a dopamine precursor) in the presence of hydrogen peroxide H₂O₂ and ascorbic acid. For in vivo applications, the authors modified the core structure of the Fe₃O₄ nanozyme as follows:

A) The authors constructed a tyrosine aptamer-based nanozyme (Apt-Fe₃O₄) to promote substrate capture despite the crowded intracellular macromolecular environment.

B) To specifically target pathological neurons in the brain and to avoid uptake in other neurons (potentially leading to side effects) the investigators performed hybridization between the Apt-Fe₃O₄ and a block strand containing α -synuclein (SNCA) mRNA antisense oligonucleotide and transferrin receptor (TfR) aptamer under two assumptions:

- the resulting molecular construct would cross the blood brain barrier in vivo and enter the intracellular neuronal compartment through receptor-mediated transport implicating the transferrin receptor

- the catalytic activity of the nanozyme in the cell would be enabled by strand displacement triggered by high intracellular SNCA mRNA levels that are specifically found in pathological neurons.

C) Furthermore, the aforementioned strand displacement reactions is expected to promote gene therapy by the release of the SNCA mRNA antisense oligonucleotide, opening the possibility of a disease-modifying effect.

These hypotheses seem to be tested in well-designed complex experiments that included appropriate controls. The experimental procedures could be divided into 3 parts:

- construction and characterization of FNA-Fe₃O₄
- in vivo cellular experiments showed that the intracellular dopamine content gradually raised & reached a plateau within 8 h without evidence for cellular toxicity
- in vivo animal experiments after injection in the tail vein in wild C57BL/6N mice, and in Thy1-SNCA transgenic mice (which carries human wild type SNCA driven by the murine thymus cell antigen 1 (Thy1) promoter) used as normal and PD animal models, respectively, provided evidence for continuous in situ dopa production as shown by striatal microdialysis and mitigation of aberrant SNCA overexpression in transgenic mice. At the behavioural level, this was associated with improved motor performance in the pole climbing test and the wire suspension test. Biodistribution analyses showed that FNA-Fe₃O₄ accumulated in large amounts in the liver, and in small amounts in the kidney, lung and spleen. Cell viability test results did not show evidence for FNA-Fe₃O₄ cytotoxicity.

Despite the complexity of the proposed methodological approach, the manuscript is well written with appropriate figures to support the results. The methods are sound although I'm not an expert in molecular biology. Overall, if the methods could be reproduced in the future, the approach hold great promise for the treatment of Parkinson's disease. One important aspect to address in the near future concerns detailed toxicity assessment of the nanozyme in normal and pathological individuals.

My recommendation is: accept the manuscript with minor comments on potential whole body toxicity to be addressed in the discussion section.

Response:

Thanks for the reviewer's positive comments and suggestions after careful reading. The replies are as follows:

During the literature investigation before the experiment, we learned that Fe₃O₄ nanoparticles were approved by the U.S. Food and Drug Administration for use in biomedical fields and have been widely used in magnetic resonance imaging, targeted drug carriers and tissue engineering. Nucleic acids, as one of the components of living body, have been widely recognized for their biocompatibility. Therefore, FNA-Fe₃O₄ constructed by Fe₃O₄ nanoparticles and functional nucleic acids should have good biosafety in theory.

In the original manuscript, we characterized the Dopa contents and dopamine contents in peripheral tissues (Supplementary Fig. 35 and 36), pathological sections of main organs (Supplementary Fig. 37), and body weight (Supplementary Fig. 40) of mice after intravenous administration of FNA-Fe₃O₄. The corresponding results confirmed that FNA-Fe₃O₄ had good biosafety.

In the revised manuscript, we carried out the hemolytic test of FNA-Fe₃O₄ on red blood cells (Supplementary Fig. 39) and the serum biochemical assay after intravenous administration of FNA-Fe₃O₄ (Supplementary Fig. 38). The corresponding results were added in the revised manuscript and supplemented the good biosafety of FNA-Fe₃O₄. The relevant discussion was added (Page 12, right column, lines 13-21).

Reviewer #4 (Remarks to the Author):

In this manuscript, the authors described a method to treat Parkinson's disease. They designed the functional nucleic acid-based responsive artificial enzyme FNA-Fe₃O₄ exhibited efficient tyrosine hydroxylase-mimicking activity. While this work appears to be of some interest, I suggest rejecting this manuscript at least in its current form. The main reasons are: 1) the construction of DNA-modified Fe₃O₄ is not novel; 2) it's surprising that DNA aptamers survive in blood circulation and then penetration BBB to reach the brain.

Response:

We appreciate for the reviewer's careful reviews and comments. The responses to the reviewer's two main viewpoints are herein clarified as follows:

(1) Reply to the question “the construction of DNA-modified Fe₃O₄ is not novel”: The innovation of our work is to propose a novel treatment strategy for Parkinson's disease by using the intrinsic properties of functional nucleic acids and Fe₃O₄. DNA-modified Fe₃O₄ has indeed been reported to be used in various fields, such as analysis and detection, MRI imaging, tumor treatment and so on. However, functional nucleic acids and Fe₃O₄ are multifunctional components, and their combination can produce countless possibilities. In this work, the catalytic ability of Fe₃O₄ nanoparticles on tyrosine hydroxylation, the capture of endogenous substrates by tyrosine aptamers, the gene therapy of antisense oligonucleotides and the brain-targeting of transferrin receptor (TfR) aptamers were used to construct the responsive artificial enzyme for fluorescence imaging and synergistic treatment of Parkinson's disease (PD). This work not only expands the catalytic properties of Fe₃O₄ nanoparticles, but also puts forward the in situ Dopa supply treatment according to the pathological characteristics of PD. Therefore, this work is of great significance for the frontier research of PD.

(2) Reply to the question “it's surprising that DNA aptamers survive in blood circulation and then penetration BBB to reach the brain”: During the literature investigation before the experiment, we found that DNA aptamers (specially modified on nanomaterials) were relatively stable in the process of blood circulation. It was reported that TfR aptamer modified nanoparticles could penetrate BBB to reach the brain in vivo, and the aptamer structure and brain-targeting ability remained stable during blood circulation (*J. Am. Chem. Soc.* 2020 (142), 3862-3872. DOI: 10.1021/jacs.9b11490; *Nano Res.* 2023 (16), 735-745. DOI: 10.1007/s12274-022-4402-7; *Small* 2022 (18), 2203448. DOI: 10.1002/sml.202203448).

In the original manuscript, we proved that FNA-Fe₃O₄ could be gradually enriched in mice brain by living fluorescence imaging and ex vivo fluorescence organ images of FNA-Fe₃O₄ injected mice (Supplementary Fig. 27 and 29 in the revised manuscript).

To further prove the stability of functional nucleic acids structure on FNA-Fe₃O₄, two stability tests were carried out and added to the revised manuscript. The first one is the penetration ability of whole blood incubated FNA-Fe₃O₄ toward BBB cell model. As shown in Supplementary Fig. 16, FNA-Fe₃O₄ still retained its ability to penetrate the in vitro BBB model after whole blood co-incubation. The relevant discussion was added (Page 8, left column, lines 6-13). The second one is the molecular stability of functional nucleic acids in mouse whole blood incubated FNA-Fe₃O₄. After 4 h blood incubation in vitro, FNA-Fe₃O₄ was magnetically separated and the strand was displaced by SNCA mRNA for mass spectrometry analysis. As shown in Supplementary Fig. 26, the mass spectra of block strand in FNA-Fe₃O₄ remained

stable after 4 h blood incubation in vitro, which enabled FNA-Fe₃O₄ to maintain its functional nucleic acid structure during blood circulation for brain targeting. The relevant discussion was added (Page 10, right column, lines 15-20). These results proved that TfR aptamer could survive in blood circulation and enhance the enrichment of FNA-Fe₃O₄ in brain.

I also request the authors to consider the following concerns.

1. The authors claimed that “Fe₃O₄ nanozymes were covalently conjugated with the 5'-terminal amino-modified tyrosine aptamer through EDC/NHS mediated amide condensation reaction to construct the tyrosine aptamer-based artificial enzyme Apt-Fe₃O₄” in page 5. Does DNA modification with different densities affect the catalytic efficiency of artificial enzyme? The authors needed to calculate how many strands of DNA were modified on each nanoparticle.

Response:

Thanks for the reviewer's comments and suggestions. Different DNA modification densities do affect the catalytic performance of artificial enzymes. However, in this study, we used the same experimental method for different DNA modified Fe₃O₄ (Apt-Fe₃O₄, Ran-Fe₃O₄ and Cat-Fe₃O₄) to ensure that the DNA modification densities on different samples were consistent. Therefore, the comparison of nanomaterial properties in this study was based on the same DNA modification density.

In the revised manuscript, the number of DNA strands modified on each nanoparticle was calculated as about 25. The relevant discussion was added (Page 5, right column, lines 21-23) and the calculation method was described (Supplementary Page 6, lines 9-10).

2. The authors described that “we proved that Fe₃O₄ could generate Fe(IV)=O active intermediates” in page 11. The authors should provide more data to support this claim.

Response:

Thanks for the reviewer's constructive suggestion. Following the suggestion, Fe(IV)=O in the reaction system was qualitatively detected and the results were added to the revised manuscript. Dimethyl sulfoxide (DMSO) is used as a probe for Fe(IV)=O, which reacts with Fe(IV)=O through oxygen atom transfer to produce dimethyl sulfone (MSM) (*Angew. Chem. Int. Ed.* 2005 (44), 6871-6874. DOI: 10.1002/anie.200502686). According to the GC-MS analysis results in Fig. 2i in the revised manuscript, the peak of MSM gradually increased with reaction time, suggesting the occurrence of Fe(IV)=O. The corresponding discussion was added (Page 5, left column, lines 38-45).

3. The level of dopamine secreted by PD cells treated with FNA-Fe₃O₄ is much higher than normal cells in Fig. 5c. Whether the treatment will affect the homeostasis of cells?

Response:

Thanks to the reviewer's comment. Treatment with the responsive artificial enzyme FNA-Fe₃O₄ will not affect the dopamine homeostasis of cells.

(1) In the cell model of Fig. 5c, FNA-Fe₃O₄ could increase dopamine levels in PD neurons. While in the actual animal model, the dopamine levels of degenerated neurons in PD were lower than normal (as shown in the comparison between the green curve and the black curve in Fig. 6c). Therefore, the increase in dopamine level was beneficial to restore dopamine homeostasis in these cells.

(2) The responsive artificial enzyme FNA-Fe₃O₄ in this study was responsive to the overexpression of SNCA mRNA in PD neurons. FNA-Fe₃O₄ exhibited a high catalytic activity for tyrosine hydroxylation in PD neurons (red curve in Fig. 5c), but a lower catalytic activity in normal neurons (black curve in Fig. 5c). Therefore, FNA-Fe₃O₄ would not affect the homeostasis of normal neurons. The relevant discussion has been mentioned in the manuscript (Page 9, right column, lines 3-5; Page 10, left column, lines 1-8).

4. In Fig. 6g, why was there almost no therapeutic effect in group ASO-Fe₃O₄?

Response:

Thanks to the reviewer's comment. As shown in Fig. 6g, h, i, the therapeutic effect of ASO-Fe₃O₄ was slightly lower than that of Cat-Fe₃O₄. This is because that the contribution of antisense oligonucleotide therapy (ASO-Fe₃O₄) is to alleviate the further deterioration of PD, but cannot reverse the pathological process of PD. While in situ Dopa supply (Cat-Fe₃O₄) can directly act on dopaminergic nervous system, thus showing more obvious therapeutic effects. However, the combination of antisense oligonucleotide therapy and in situ Dopa supply exhibited better synergistic therapeutic effects, just as the FNA-Fe₃O₄ treated mice exhibited better performance than monotherapy. This phenomenon could be attributed to the fact that antisense oligonucleotide therapy protected mice from further deterioration after their motor performance was partially restored.

In the revised manuscript, we discussed the contribution of Cat-Fe₃O₄ and ASO-Fe₃O₄ to the treatment of PD in behavioral tests (Page 12, left column, lines 35-41, 46-48).

5. In addition to BV2 cells, the authors should also test the cytotoxicity of the artificial enzyme to other related cells.

Response:

Thanks for the reviewer's thoughtful suggestion. Following the suggestion, two cell compatibility experiments were carried out and added to the revised manuscript. The first is the cytotoxicity test of the artificial enzymes on the neuron model human neuroblastoma SH-SY5Y cells. As shown in Supplementary Fig. 20, FNA-Fe₃O₄, Cat-Fe₃O₄ and ASO-Fe₃O₄ did not cause obvious death of SH-SY5Y cells, indicating good biocompatibility with neurons. The relevant discussion was added (Page 8, right column, lines 17-19). The second is the hemolysis test of FNA-Fe₃O₄ to red blood cells. As shown in Supplementary Fig. 39, negligible hemolytic activity of FNA-Fe₃O₄ when incubated with red blood cells was shown, which indicated good hemocompatibility. The relevant discussion was added (Page 12, right column, lines 18-21). These results proved that the artificial enzyme FNA-Fe₃O₄ had good biocompatibility.

6. The format of references needs to be uniform.

Response:

Thanks for the reviewer's comment. In ref. 48 of the original manuscript, we quoted a book, which was used to provide reference for stereotactic orientation of striatal microdialysis. In the revised manuscript, the ref. 50 (that is, ref. 48 in the original manuscript) was changed into a more accurate literature and the format was updated. And the format of all references were uniformed.

Reviewers' Comments:

Reviewer #1:

Remarks to the Author:

The authors have made significant changes, and replied the comments that the reviewers addressed. It can be accepted for publication.

Reviewer #2:

Remarks to the Author:

The authors have addressed all my concerns. It can be accepted at current form.

Reviewer #4:

Remarks to the Author:

I think this revision is satisfactory.

Response to Reviewers' Comments

Reviewer #1 (Remarks to the Author):

The authors have made significant changes, and replied the comments that the reviewers addressed. It can be accepted for publication.

Response:

Thanks for the reviewer's positive comments.

Reviewer #2 (Remarks to the Author):

The authors have addressed all my concerns. It can be accepted at current form.

Response:

Thanks for the reviewer's positive comments.

Reviewer #4 (Remarks to the Author):

I think this revision is satisfactory.

Response:

Thanks for the reviewer's positive comments.